# EVADE-BENCH: MULTIMODAL BENCHMARK FOR EVASIVE CONTENT DETECTION IN E-COMMERCE APPLICATIONS

## ABSTRACT

E-commerce platforms increasingly rely on Large Language Models (LLMs) and Vision–Language Models (VLMs) to detect illicit or misleading product content. However, these models remain vulnerable to *evasive content*: inputs (text or images) that superficially comply with platform policies while covertly conveying prohibited claims. Unlike traditional adversarial attacks that induce overt failures, evasive content exploits ambiguity and context, making it far harder to detect. Existing robustness benchmarks provide little guidance for this high-stakes, real-world challenge. We introduce **EVADE-Bench**, the first expert-curated, Chinese, multimodal benchmark specifically designed to evaluate foundation models on evasive content detection in e-commerce. The dataset contains 2,833 annotated text samples and 13,961 annotated images spanning six categories, including body shaping, height growth, health supplements, and others. Two complementary tasks assess distinct capabilities: *Single-Violation*, which probes fine-grained reasoning under short prompts, and *All-in-One*, which tests long-context reasoning by merging overlapping policy rules into unified instructions. Our benchmarking of 26 mainstream LLMs and VLMs reveals that even state-of-the-art models frequently misclassify evasive samples. By releasing EVADE-Bench, we provide the first rigorous standard for evaluating evasive-content detection, expose fundamental limitations in current multimodal reasoning, and lay the groundwork for safer and more transparent content moderation systems in e-commerce.

## 1 INTRODUCTION

In recent years, Large Language Models (LLMs) (Zhao et al., 2025; Xiao et al., 2025) and Vision Language Models (VLMs) (Li et al., 2023; Liang et al., 2024) have made significant progress across various fields. These models have gained widespread attention for their applications in natural language processing, image recognition, and multimodal tasks, and continue to drive technological advancements across industries (Matarazzo & Torlone, 2025; Radford et al., 2021; OpenAI et al., 2024b). Particularly in the e-commerce domain, these models have been extensively applied to tasks such as product search, recommendation, and content moderation (Ren et al., 2024; Jiang et al., 2024). However, when confronted with the task of Evasive Content Detection (ECD)—identifying text or image content that has been deliberately altered to circumvent platform rules while still conveying misleading information—they exhibit significant limitations in performance.

The task of ECD represents an adversarial dynamic between sellers and platform policies, which differs from traditional adversarial attacks (Zou et al., 2023; Hackett et al., 2025). Conventional adversarial attacks typically manipulate inputs (e.g. tiny pixel noise or prompt injections) to induce harmful or incorrect model outputs (Chowdhury et al., 2024; Liu et al., 2024a). In contrast, the adversarial nature of ECD lies at the data level. For instance, sellers may obscure prohibited health claims using euphemistic language or crop images to conceal policy-violating details. When such content escapes detection, platforms face regulatory fines, lawsuits, fraudulent transactions and reputational damage—economic losses that rival technical security breaches (Palen-Michel et al., 2024).

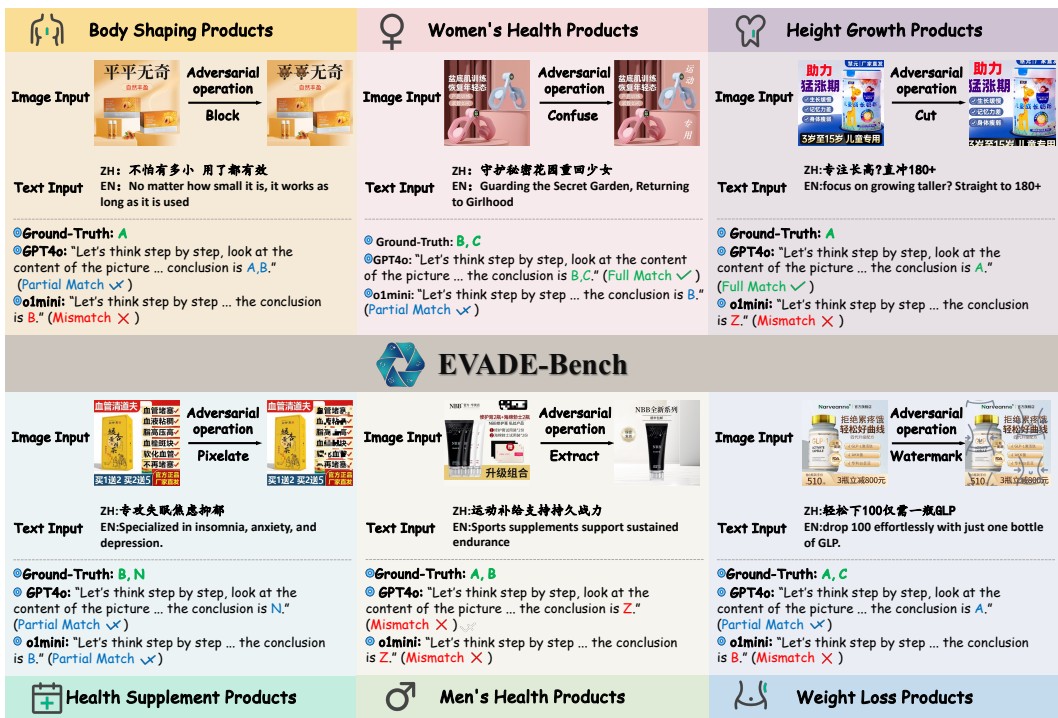

Figure 1: Illustrations of EVADE-Bench Samples.

Merchants often try to enhance product sales by exaggerating claims, using ambiguous language, or distorting images to attract consumers and bypass platform content review. In contrast, platforms must ensure fair transactions and must identify and regulate misleading product promotions as effectively as possible. In this context, merchants continuously attempt to exploit loopholes in advertising laws by using misspellings, slang, emojis, and mosaics to modify text and images, avoiding the platform's automatic scrutiny. Thus, adversarial issues in the e-commerce sector are even more pronounced.

Despite these violations, evasive content detection remains underexplored. Current models are particularly limited for two key reasons. First, LLMs and VLMs continue to hallucinate (Huang et al., 2025; Bai et al., 2025b), mis-follow instructions (Lou et al., 2024; Murthy et al., 2025), and struggle with long or multimodal contexts (Wang et al., 2024; Chen et al., 2024), making it difficult to recognize nuanced deception. Second, real-world e-commerce regulations evolve rapidly, and human annotations are often inconsistent, leading to noisy and ambiguous training data. Together, these factors hinder the development and evaluation of models capable of trustworthy moderation.

To address this gap, we introduce the **Eva**sive Content **D**etection in **E**-Commerce **Bench**mark (**EVADE-Bench**), a Chinese multimodal benchmark specifically designed for evaluating how well LLMs and VLMs detect evasive content in real-world e-commerce scenarios. Every sample is iteratively annotated by domain experts to ensure accurate and consistent ground truth. EVADE-Bench contains 2,833 text samples and 13,961 product images, collected from six high-stake product categories where deceptive content is most prevalent: weight loss, diseases, height growth, body shaping, female care, and male care. Each sample is iteratively annotated by domain experts to ensure its accuracy and reliability. Furthermore, we design two tasks for EVADE-Bench: Single-Violation and All-in-One. The former evaluates a model's ability to perform rapid and fine-grained judgments within in a narrowly scoped context, while the latter explores its capacity for reasoning under longer and more complex rule-intensive conditions.

EVADE-Bench offers two evaluation tracks to assess both a model's ability to quickly identify specific violations and its robustness when faced with complex rules. We conduct a comprehensive evaluation of 26 mainstream LLMs and VLMs on EVADE-Bench, analyzing their performance on this challenging task.

**Our key contributions are as follows:**

1. We release EVADE-Bench, the first expert-curated, Chinese multimodal dataset tailored for evasive content detection in e-commerce. The benchmark features two evaluation tracks—Single-Violation and All-in-One—that probe distinct reasoning capabilities under varying policy contexts.

2. We show that clearer rule categorization significantly improves model consistency and reduces false predictions, highlighting the importance of benchmark design for reliable evaluation.

3. We benchmark 26 open- and closed-source LLMs and VLMs, providing the first systematic baseline for this underexplored yet high-impact problem. We then examined common errors made by these models on EVADE-Bench, analyzed potential issues in multimodal large models, and explored the feasibility of using Multi-Agent decomposition for multimodal reasoning to improve model accuracy.

Through these contributions, we aim to catalyze research in adversarial and evasive content detection, support the development of safer and more trustworthy moderation systems, and advance the field of robust multimodal reasoning in high-stakes commercial settings.

## 2 RELATED WORK

**Evasive and Obfuscated Content Detection** Recent work on online safety highlights adversarial and obfuscated content detection in hate speech and cyberbullying. SWE2 (Mou et al., 2020) enhances robustness to lexical attacks by combining word- and subword-level features, while an LSTM-based model with correction mechanisms improves resilience to deceptive cyberbullying patterns (Azumah et al., 2024). Autoregressive models have also been used to craft graded adversarial examples for implicit hate detection (Ocampo et al., 2023), introducing a "build-it, break-it, fix-it" retraining loop that boosts model robustness to nuanced, context-sensitive abuse.

**Deceptive or Policy-Violating Content in E-commerce** The safety and robustness of VLMs have gained attention with specialized benchmarks. The Hateful Memes challenge (Kiela et al., 2021) pioneered rigorous multimodal evaluation by using subtle hateful content contrasted with benign distractors, discouraging unimodal shortcuts. Later benchmarks like MM-SafetyBench (Liu et al., 2024b) used 5,040 adversarial image-text pairs to show that even aligned models are vulnerable to malicious prompts. MMSafeAware (Wang et al., 2025) found GPT-4V misclassified over one-third of unsafe and over half of safe inputs, exposing poor safety awareness across 29 threats. VLD-Bench (Raza et al., 2025) evaluated 31,000 news-image pairs and showed that adding visual context improves disinformation detection by 5–35% and enhances compliance monitoring.

## 3 THE EVADE-BENCH

We introduce EVADE-Bench, a multimodal benchmark grounded in Chinese advertising law regulations, designed to evaluate whether current LLMs and VLMs can effectively identify evasive content in real-world e-commerce settings. EVADE-Bench integrates both textual and visual inputs, requiring models to jointly comprehend and reason over multimodal data in accordance with explicit policy guidelines.

### 3.1 OVERVIEW OF THE EVADE-BENCH

This benchmark comprises 2,833 text samples and 13,961 images, all collected from real-world e-commerce platforms. Each instance has been manually annotated by domain experts with deep familiarity in advertising law, ensuring high-quality and regulation-compliant labels.

For each sample (image or text) in EVADE-Bench, we provide a corresponding text prompt that contains all rules of the current data type. We construct a multimodal input pair by concatenating each image with its prompt for VLMs' reasoning, and construct a pure text input pair by concatenating each text with its prompt for LLMs' reasoning. The detailed prompts for each data type can

Table 1: The data distribution of EVADE-Bench and the prompt length corresponding to each violation category.

| Data-Type | Text-Count | Image-Count | Prompt-Len |
|---|---|---|---|
| Body Shaping | 202 | 2,134 | 614 |
| Women's Health | 211 | 1,295 | 652 |
| Height Growth | 553 | 3,424 | 953 |
| Men's Health | 652 | 1,738 | 1,123 |
| Weight Loss | 442 | 1,203 | 1,364 |
| Health Supplement | 773 | 4,167 | 3,379 |
| Overall | 2,833 | 13,961 | / |

be found in **Appendix G**. For a more fine-grained distribution of samples in EVADE-Bench, please refer to **Appendix D**.

Images present a particularly challenging modality, as they often embed both textual claims and visual cues. Thus, image-based reasoning not only requires the VLMs to correctly extract embedded text via OCR but also to interpret visual elements in conjunction with text prompt. This dual-modality reasoning is essential for detecting subtle forms of evasion, such as cropped disclaimers or euphemistic imagery. At the same time, texts present unique challenges for LLMs due to intentionally created variations such as missing keywords, homophones, and typos.

EVADE-Bench contains complex logical operations and requires models to correctly identify the content of given information. The model needs to correctly understand the **Four-Step Logic System** to provide the right answer based on previously identified content information. For detailed information about the four-step logic system, please refer to **Appendix F**.

### 3.2 DATA COLLECTION AND RULE FORMULATION

To construct a diverse and regulation-aligned benchmark, we collected 25,380 raw texts and 48,000 raw images from six e-commerce sub-domains (e.g., body shaping, height growth). Through collaboration with advertising experts, we designed six rules based on Chinese advertising law. While expert human annotation provides high-fidelity labels, it is not immune to inconsistencies due to fatigue, subjectivity, or ambiguous cases. To ensure dataset uniqueness, diversity, and quality, we introduced a multi-stage pipeline for collecting challenging samples and generating ground-truth from raw images and texts.

First, we employed LLMs and VLMs of various sizes to automatically generate a high-quality dataset from raw images and texts. The detailed pipeline is presented in **Algorithm 1**, with validation models listed in **Table 14**. A sample is considered simple and excluded from $D_{unlabeled}$ if all models, including the smallest one, produce consistent predictions, highlighting the challenging nature of $D_{unlabeled}$. At this stage, the dataset temporarily lacks ground-truth labels.

Next, trained expert annotators labeled each instance and compared their annotations with predictions from the three best LLMs (GPT-o1mini, DeepSeek-R1, QwenMax) and VLMs (GPT-4o, Claude-3.7, Gemini-2.5-Pro) to identify inconsistent cases. Through iterative comparison and annotation, we ultimately generated ground-truth labels for each instance in the dataset. The detailed pipeline is presented in **Algorithm 2**.

To prevent dataset contamination and maintain unbiased evaluation, we ensure that experts perform their annotations without access to model predictions.

## 4 EXPERIMENT

**Baselines** All LLMs and VLMs used in our experiments are listed in **Appendix H**. The models include both open-source and closed-source representative models. We conducted a Human Baseline Experiment with three independent domain experts to annotate all texts and images from EVADE-Bench and calculated Krippendorff's Alpha to analyze human annotation quality. Due to its 4K context length limitation, DeepSeek-VL2-27B was only used for the Single-Violation task.

**Tasks**  We conducted comprehensive experiments on EVADE-Bench using a suite of both open-source and closed-source LLMs and VLMs. The evaluation is structured around two core tasks: Single-Violation and All-in-One, designed to probe distinct capabilities in complex reasoning.

**Single-Violation**  The task evaluates model performance across six distinct product categories using short, domain-specific prompts (e.g., 202 texts and 2,134 images assessed with a 614-token prompt for body shaping products). It tests the model's fine-grained reasoning ability within narrowly defined contexts. However, the task faces a key challenge: semantic overlaps between categories (e.g., "weight loss" products often claim health benefits, while "health improvement" products frequently emphasize weight management as a health outcome) can confuse the model's decision-making. To simulate more complex, classification-dense scenarios, we propose the All-in-One task.

**All-in-One**  In the All-in-One task, we unify prompts across six violation types into a single instruction, expanding prompt length from 1K to 7K tokens and increasing classification labels from an average of 5 (in Single-Violation) to 26 distinct regulatory categories. We merge semantically overlapping rules to reduce ambiguity. This means increasing both the context length and the number of classifications during model inference, but reducing the possibility of confusion.

To assess model generalization in adversarial e-commerce scenarios, we define two sub-tasks: the Simplified Instruction task and the Detailed Instruction task. The former resembles zero-shot inference, while the latter simulates few-shot reasoning.

1. **Simplified Instruction** refers to the approach where, in the input prompt, we avoid introducing any examples except for the necessary definition. The purpose of this approach is to allow models for free reasoning based on the prompt, in order to explore the upper and lower bounds of its performance.

2. **Detailed Instruction** refers to the approach where, in the input prompt, we not only include the necessary definition but also introduce positive and negative examples. The purpose of this approach is to constrain the model's free-form generation through detailed examples and descriptions, thereby stabilizing its performance.

We employ the prompt *Let's think step by step* to guide the model's systematic reasoning process across all experiments.

## 4.1 EVALUATION INDICATORS

Since our task requires models to analyze samples from EVADE-Bench and provide a final classification result based on the prompt, we evaluate model performance using two metrics: partial accuracy and full accuracy. **Full Accuracy** requires the model's final classification result to exactly match the ground truth, while **Partial Accuracy** is achieved when the model's classification result overlaps with at least one category in the ground truth.

$$\text{Acc}_{full} = \frac{1}{N} \sum_{i=1}^{N} \mathbb{I}(C_i = G_i) \quad \Big| \quad \text{Acc}_{part} = \frac{1}{N} \sum_{i=1}^{N} \mathbb{I}(C_i \cap G_i \neq \varnothing)$$

where $N$ is the total number of samples. For each sample $i$, $C_i$ represents the set of predicted items and $G_i$ represents the set of ground truth items. The term $\mathbb{I}(\cdot)$ is an indicator function that evaluates to 1 if true, and 0 otherwise.

In real e-commerce scenarios, these two metrics serve different purposes. Full Accuracy measures the degree of alignment between model and human understanding of a sample. Partial Accuracy, on the other hand, addresses a practical challenge in e-commerce where categories lack the clear boundaries found in mathematical or coding tasks. When dealing with metaphorical content, semantic overlaps between classifications are inevitable. In such cases, pursuing extremely high Full Accuracy might lead to model overfitting. In practice, e-commerce moderators focus more on identifying rule violations rather than categorizing the specific type of violation.

Therefore, while Full Accuracy aligns better with mainstream benchmark metrics, Partial Accuracy is more commonly used in practical e-commerce applications.

# 5 MAIN RESULTS

## 5.1 SINGLE-VIOLATION RESULTS

For VLMs processing image-text pair inputs, as shown in **Table 2**, the closed-source models Claude-3.7-sonnet and GPT-4o achieve the highest overall accuracy across all six categories. Among open-source VLMs, Qwen2.5-VL-72B demonstrates the best performance, showing competitive results despite its open-source nature. In contrast, DeepSeek-VL2-27B shows the weakest performance, significantly lagging behind even smaller models such as 8B-parameter VLMs, highlighting the impact of model architecture and training strategies rather than model size alone.

Table 2: All model overall performance on Single-Violation of EVADE-Bench.

| LLMs | Partial Acc. / Full Acc. | VLMs | Partial Acc. / Full Acc. |
|---|---|---|---|
| *Open Source* | | | |
| Llama-8B | 35.62 / 20.93 | MiniCPM-V2.6 | 44.15 / 12.37 |
| LLama-70B | 38.55 / 26.23 | InternVL3-8B | 42.48 / 18.87 |
| Qwen-7B | 38.48 / 27.18 | InternVL3-14B | 51.20 / 23.23 |
| Qwen-14B | 45.39 / 28.70 | InternVL3-38B | 49.19 / 21.40 |
| Qwen-32B | 46.56 / **29.69** | DeepSeek-VL2 | 29.12 / 12.24 |
| Qwen-72B | 49.21 / 27.85 | Qwen-VL-7B | 44.52 / 19.96 |
| DeepSeek-V3 | 51.85 / 28.77 | Qwen-VL-32B | 52.39 / 22.57 |
| DeepSeek-R1 | **54.64** / 25.45 | Qwen-VL-72B | **57.63** / **25.05** |
| *Close Source* | | | |
| GPT-o1mini | 49.28 / 29.72 | GPT-4o | 58.47 / **26.96** |
| GPT-4.1 | **52.74** / **31.59** | Claude-3.7 | **58.79** / 23.42 |
| Qwen-max | 48.29 / 31.27 | Qwen-VL-max | 53.38 / 25.60 |
| | | Gemini-2.5-pro | 52.44 / 22.14 |
| *Human Performance* | | | |
| Human on Text | 69.34 / 57.03 | Human on Image | 69.20 / 51.41 |

For LLMs processing pure text inputs, the LLaMA series struggles significantly due to limited Chinese language capabilities. Even the 70B version of LLaMA performs only at par with the 7B version of the Qwen series, revealing a critical limitation in multilingual robustness for otherwise powerful models.

Table 3: Human annotation on EVADE-Bench's text and image tasks achieved Krippendorff's Alpha scores of 0.67~0.8, demonstrating acceptable inter-annotator agreement.

| | Overall | Body. | Women. | Height. | Men. | Weight. | Health. |
|---|---|---|---|---|---|---|---|
| Human on Image | 0.7513 | 0.5961 | 0.5737 | 0.7638 | 0.6903 | 0.6502 | 0.7148 |
| Human on Text | 0.6867 | 0.2472 | 0.3810 | 0.6137 | 0.6716 | 0.6719 | 0.6125 |

The Krippendorff's Alpha scores (0.67 ~ 0.8) indicate substantial agreement among experts, particularly on images ($\geqslant$0.75 overall), proving that despite the task's complexity, the annotation rules are well-defined and consistent for humans, which dispels concerns about label noise. Although models perform competitively on certain categories, humans still outperform all LLMs and VLMs overall in understanding evasive text and images (Partial Acc.) and also surpass current models in precision (Full Acc.). Notably, on EVADE-Bench's most challenging data (Health Supplement), humans show an even larger performance gap over all models (at **Appendix E**). This demonstrates that humans have strong generalization ability with evasive content, maintaining stable reasoning performance across different types of text and images, unlike LLMs and VLMs which exhibit high accuracy fluctuations across data types.

A critical observation across all models is the significant gap between partial accuracy and full accuracy, often exceeding 10%. This gap shows how models struggle to achieve complete under-

Table 4: Qwen3 is a newly released series of LLMs that integrates both *thinking* and *non-thinking* modes within a single architecture. However, its largest variant, Qwen3-235B-A22B, still underperforms compared to DeepSeek-R1-671B and DeepSeek-V3-671B. Surprisingly, enabling the thinking mode yields minimal improvements and even leads to performance degradation in some models, such as Qwen3-32B.

| Model-Flat-Infer | Partial Acc. | Full Acc. | Model-Think-Infer | Partial Acc. | Full Acc. |
|---|---|---|---|---|---|
| Qwen-3-32B | 48.71 | 26.16 | Qwen-3-32Bb | $47.86_{(-0.85)}$ | $24.71_{(-1.45)}$ |
| Qwen-3-30B-A3B | 46.70 | 26.44 | Qwen-3-30B-A3B | $47.51_{(+0.81)}$ | $\mathbf{27.64}_{(+1.20)}$ |
| Qwen-3-235B-A22B | **49.77** | **27.46** | Qwen-3-235B-A22B | $\mathbf{50.02}_{(+0.25)}$ | $27.50_{(+0.04)}$ |

standing: while they can catch basic meanings or simple features, they often miss important details in meaning or images, especially when dealing with indirect language, hidden meanings, or misleading images. While closed-source models generally perform better than open-source ones, there are clear exceptions. For instance, DeepSeek-R1 shows strong results in text understanding, even performing better than some closed-source models, while Qwen2.5-VL-72B shows good ability in image understanding, though not quite reaching the level of GPT-4o or Claude-3.7-sonnet.

## 5.2 ALL-IN-ONE RESULTS

We observe substantial performance improvements across models, particularly in smaller LLMs and VLMs, as shown in **Table 5**. Without altering the input data, merely combining overlapping classifications significantly enhances the models' reasoning capabilities. Despite a sixfold increase in prompt length (compared to the Single-Violation setting) and an expansion from a few classifications to 26, models generally demonstrate improved rather than degraded performance. This suggests that the clarity of boundaries between classifications poses a more significant challenge for models than the context length or the number of classifications. Notably, the performance gap among LLMs narrows significantly in the All-in-One setting. Additionally, the difference between partial and full accuracy decreases dramatically, dropping from over 10% in the Single-Violation setting to approximately 5% in All-in-One.

Table 5: The performance of all models on the All-in-One task of EVADE-Bench. "Simp. Instr." represents Simplified Instruction, while "Det. Instr." represents Detailed Instruction.

| LLMs | (Part / Full) Acc. | | VLMs | (Part / Full) Acc. | |
|---|---|---|---|---|---|
| | Simp. Instr. | Det. Instr. | | Simp. Instr. | Det. Instr. |
| *Open Source* | | | | | |
| Llama3.1-8B | 41.94 / 36.47 | 37.84 / 34.44 | MiniCPM-V2.6 | 46.70 / 43.22 | 39.60 / 36.93 |
| LLama3.1-70B | 47.81 / 44.53 | 47.00 / 44.60 | InternVL3-8B | 56.26 / 53.90 | 56.97 / 54.17 |
| Qwen2.5-7B | 51.68 / 49.52 | 53.18 / 50.60 | InternVL3-14B | 60.63 / 56.83 | 60.67 / 57.41 |
| Qwen2.5-14B | 53.14 / 48.32 | 55.23 / 51.20 | InternVL3-38B | 60.62 / 57.35 | 61.36 / 58.34 |
| Qwen2.5-32B | 55.12 / **51.20** | 53.55 / 49.95 | Qwen2.5VL-7B | 53.15 / 51.49 | 53.39 / 51.20 |
| Qwen2.5-72B | 56.93 / 51.06 | 55.59 / 50.05 | Qwen2.5VL-32B | 62.04 / 58.55 | 61.73 / 59.17 |
| DeepSeek-V3 | 56.58 / 50.62 | **58.79 / 52.65** | Qwen2.5VL-72B | **64.25 / 59.09** | 63.85 / 59.44 |
| DeepSeek-R1 | **58.25** / 49.35 | 58.69 / 50.37 | | | |
| *Close Source* | | | | | |
| GPT-o1mini | 56.69 / 51.15 | 54.17 / 48.76 | GPT4o-0806 | 64.14 / 58.05 | 65.12 / **59.58** |
| GPT4.1-0414 | **59.16** / 53.79 | **59.64 / 53.67** | Claude-3.7 | 64.58 / 56.83 | 63.05 / 56.05 |
| Qwen-max | 58.35 / **54.68** | 56.48 / 52.88 | QwenVL-max | 63.58 / **59.24** | 63.31 / 59.50 |
| | | | Gemini2.5-pro | **70.57** / 54.45 | **70.43** / 51.94 |

Interestingly, while high-performing LLMs—such as DeepSeek-R1, DeepSeek-V3, and the GPT series—can achieve notable improvements in full accuracy, their gains in partial accuracy under the All-in-One are relatively modest.

This may be attributed to ceiling effects or performance saturation that limit further improvements. In contrast, previously underperforming models show improvements across both metrics. For example, Qwen2.5-7B improves its partial accuracy from 38.48% to 53.18% and full accuracy from 27.18% to 50.60%, achieving over 10% gains in both metrics. Similarly, Llama-3.1-70B, despite its limited proficiency in Chinese, improves from 38.55% to 47.00% in partial accuracy and from 26.23% to 44.60% in full accuracy.

This trend also extends to VLMs: previously underperforming models like InternVL3-8B and MiniCPM-V2.6-8B demonstrate dramatic improvements, while large-scale models continue to achieve further gains. These results affirm the effectiveness of the All-in-One setting in reducing confusion caused by overlapping classifications and reveal the potential of structured prompt engineering for regulatory tasks.

## 5.3 ANALYSIS OF THE EFFECT OF RAG AND SFT

Given that EVADE-Bench contains fewer text samples than image samples, we investigate model reasoning enhancement through different approaches: Retriever-Augmented Generation (RAG) for the text portion and Supervised Fine-tuning (SFT) for the image portion. We conducted RAG experiments on LLMs and, due to resource constraints, performed SFT experiments exclusively on Qwen-VL series models for VLMs. We first split all text samples from EVADE-Bench into document and query sets with a ratio of 2:8. For each sample in the query set, we then identified its most similar sample from the document set using text similarity metrics to enhance LLMs' reasoning performance. For the SFT experiments, we randomly split the image samples from EVADE-Bench into training (80%) and evaluation (20%) sets. The VLMs were then fine-tuned on the training set for three epochs and evaluated on the held-out test set.

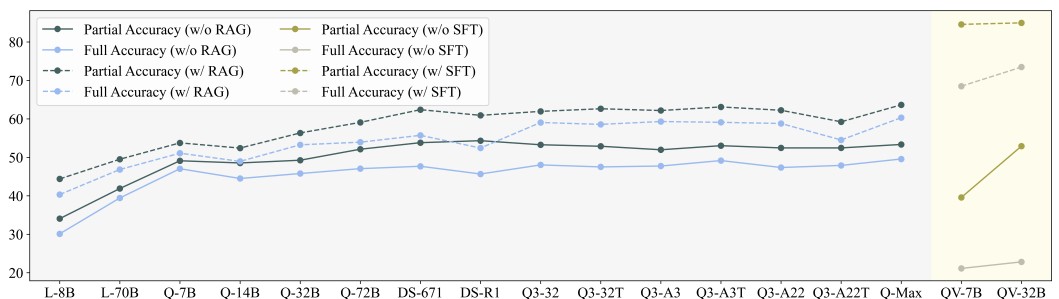

Figure 2: Comparison of LLMs and VLMs before and after the introduction of RAG and SFT. The models compared include: Llama-3.1 (abbreviated as L), Qwen2.5 (Q), DeepSeek (DS), Qwen3 (Q3; with thinking mode T), and Qwen2.5-VL (QV).

As shown in **Figure 2**, RAG effectively enhanced LLMs' reasoning performance on EVADE-Bench. Similarly, VLMs showed significant performance improvements through SFT on EVADE-Bench. These experiments demonstrated significant performance improvements for both LLMs and VLMs in Single-Violation and All-in-One tasks. Our preliminary findings indicate that RAG and SFT improve the models' precision, especially in cases involving ambiguous or metaphorical inputs, by providing semantically aligned reference materials.

## 5.4 ERROR ANALYSIS

As analyzing all error cases would be prohibitively time-consuming, we conducted our analysis on a random sample of 100 incorrect cases from three representative VLMs, scrutinizing and classifying their error types as shown in Table 6.

**Info Missing** Analyzing the performance of Qwen-2.5-VL-7B in **Table 6**, we observe that VLMs are easily disturbed when images contain missing key information or indirect implications. In such cases, VLMs must first infer the missing information before arriving at the correct answer. Moreover, VLMs need to accurately identify text through OCR before inferring the missing information,

and this process can be disrupted by image distortion—a technique merchants employ to convey information to human consumers while evading automated detection systems. This reflects a broader issue in contextual reasoning: models often lack sufficient sensitivity to metaphors, euphemisms, and dual meanings—leading to failures in detecting layered or implied semantic cues within multimodal inputs. VLMs struggle to accurately recognize embedded text in product images when it

Table 6: Randomly select the cause distribution of 100 error types. The errors can be attributed to various causes: model hallucinations during inference on non-existent elements (Hallu.); failure to locate key information (Focus.); intentionally introduced information omissions or errors in images that prevent model comprehension (Omit.); recognition failures due to image occlusion or deformation (Distort.); correct violation detection but failure to locate corresponding prompt rules (Find.); and correct violation detection but misunderstanding of prompt rules (Match.).

| Model | Hallu. | Focus. | Omit. | Distort. | Find. | Match. |
|---|---|---|---|---|---|---|
| QVL-7B | 1 | 4 | 26 | 5 | 38 | 26 |
| QVL-7B$_{sft}$ | 0 | 0 | 8 | 0 | 6 | 3 |
| QVL-72B | 0 | 0 | 26 | 6 | 36 | 32 |
| GPT-4o | 1 | 3 | 13 | 4 | 46 | 33 |

is obscured by noise, masking, or creative formatting. Even with correct region detection, these models may misread or miss critical content, leading to incorrect classification. This limitation underscores the need for enhanced OCR capabilities and improved text-visual alignment in commercial applications. By comparing the performance on the same test set before and after SFT, as shown in **Table 6**, we demonstrate that SFT significantly reduces these errors by providing VLMs with additional knowledge about violating images.

**Model Bias**   We observe an unexpected phenomenon: the stronger the model, the more likely it is to misinterpret prompt rules. This is particularly evident with GPT-4, which, despite correctly identifying key violating words, fails to properly match them to the classification criteria in the prompt—either unable to locate the corresponding violation type or misclassifying it as another type. This indicates two main challenges: (1) models struggle with backward reasoning—while they can generate key information based on prompts, they fail to match this information back to the initial prompt requirements. (2) Models' prompt interpretation shows systematic biases compared to human understanding. We hypothesize that stronger models may have higher confidence in their reasoning abilities, leading them to adhere more firmly to their own conclusions, thus amplifying deviations from human understanding and resulting in more classification errors in larger models.

**Implicit Error**   Although EVADE-Bench is a multi-classification task, it includes an implicit critical rule: Z.other (no violation) cannot co-exist with any violation classification. Each sample can be classified as either a violation type or non-violation, but not both. Models must understand this fundamental rule before making any classification decisions. Some weaker models fail to follow this constraint, incorrectly labeling items as both violation and non-violation, revealing their limited ability to understand and follow complex classification rules.

## 5.5 DECOMPOSING MULTIMODAL REASONING

We have previously analyzed the reasoning failures in current multimodal large models. We hypothesize that the fundamental cause of these errors lies in the training process of VLMs. While building upon LLMs has enabled new multimodal reasoning capabilities, it has potentially weakened the text processing abilities - a perspective validated by numerous studies(Guo et al., 2025; Zheng et al., 2025). Prism(Qiao et al., 2024) proposed decomposing multimodal reasoning into two stages: "VLM generating textual descriptions of image information" followed by "LLM performing reasoning on the text." This approach significantly improved model performance in multimodal reasoning tasks.

We think the Prism method, which decomposes multimodal reasoning into "VLM+LLM" stages, is particularly well-suited for tasks like EVADE-Bench. Building on this insight, a promising direction for future research would be to explore a decomposed conceptual approach for evasive content

Table 7: Decomposing multimodal reasoning performance comparison

| Model | Overall | Body. | Weight. | Women. | Men. | Health. | Height. |
|---|---|---|---|---|---|---|---|
| *Multimodal reasoning without decomposition as baseline* | | | | | | | |
| GPT4o | 59.61 / 29.14 | 76.82 / 32.73 | 74.11 / 08.04 | 55.20 / 40.00 | 53.72 / 35.64 | 45.64 / 27.61 | 65.26 / 28.05 |
| Claude-3.7 | 59.07 / 24.53 | 73.86 / 27.05 | 71.75 / 07.62 | 53.44 / 36.84 | 49.47 / 32.45 | 46.01 / 21.96 | 68.27 / 22.71 |
| *GPT-4o as the image description model and the below model as the reasoning model* | | | | | | | |
| GPT4.1 | **61.65** / **32.76** | 67.95 / 24.09 | 79.91 / 12.95 | **59.60** / 39.60 | **56.65** / **39.36** | 49.20 / **31.53** | 69.91 / **40.12** |
| Qwen3-235b | 61.30 / 28.64 | 67.05 / **29.09** | **82.59** / 12.50 | 56.80 / 37.20 | 54.52 / 32.98 | 48.47 / 28.34 | **71.22** / 28.49 |
| DeepSeek-R1 | 60.87 / 28.61 | **69.55** / 27.50 | 78.57 / **16.52** | 58.00 / **40.40** | 50.53 / 29.52 | **51.66** / 28.47 | 67.15 / 28.63 |
| *Claude-3.7 as image description model and the below model as the reasoning model* | | | | | | | |
| GPT4.1 | **62.55** / 30.97 | 62.95 / 22.05 | 78.12 / **14.29** | **61.60** / 41.20 | **60.64** / **44.95** | 51.66 / **25.64** | **71.51** / **37.06** |
| Qwen3-235b | 61.19 / 24.96 | 62.27 / **25.68** | **80.36** / 08.93 | 57.20 / 38.00 | 53.46 / 31.91 | **51.90** / 22.94 | 70.93 / 23.55 |
| DeepSeek-R1 | 59.86 / 24.74 | **65.68** / 25.00 | 76.79 / 14.73 | 58.80 / 39.20 | 50.00 / 30.05 | 51.29 / 20.98 | 66.57 / 24.13 |

detection. Since "generating textual descriptions of image information" doesn't involve complex logical reasoning and remains a relatively simple task, we can utilize a smaller VLM for image description generation, while employing more powerful LLMs for complex logical reasoning.

To verify the effectiveness of this approach, we conducted experiments using GPT-4o and Claude-3.7 specifically for converting images into pure text descriptions, and then employed different powerful LLMs to perform multi-class inference on the text generated by GPT-4o and Claude-3.7 alone, without access to the images themselves. As **Table 7** shows, compared to single-stage multimodal reasoning, using the same VLM to describe images and then passing the descriptions to a dedicated LLM for reasoning leads to better performance.

This is not an isolated phenomenon, this performance improvement is consistent across different VLMs and LLMs. However, the performance did not saturate, indicating that the "Description" stage itself remains challenging due to the adversarial nature of EVADE-Bench, for example, subtle visual puns. This further validates the high difficulty and value of our benchmark as a testbed for future "Restoration" capabilities. Additionally, the experiment validates our hypothesis that integrating "description + reasoning" in a single multimodal inference can weaken the model's inherent reasoning performance. This suggests that multimodal post-training may compromise the base LLM's text reasoning capabilities.

## 6 CONCLUSION

EVADE-Bench is the first Chinese-language multimodal benchmark for detecting evasive e-commerce content, covering expert-annotated texts and images across six categories. Evaluation of 26 LLMs and VLMs reveals significant performance disparities and shows that both LLMs and VLMs still fall short on EVADE-Bench. The introduction of EVADE-Bench is therefore poised to catalyze progress in this critical area. We find that merging overlapping classifications enhances reasoning and reduces the accuracy gap, despite expanding prompt length and increasing classification labels—particularly in smaller models, underscoring the importance of rule clarity. In our experiments, leveraging RAG and SFT to supply models with knowledge has yielded significant performance improvements, demonstrating the effectiveness of EVADE-Bench as an evaluation benchmark. Our analysis reveals common failure modes in EVADE-Bench, including contextual noise, obfuscated language, and OCR errors, indicating weaknesses in semantic understanding and vision-language alignment, and explored the feasibility of decomposing "single multimodal inference" into "description before inference" through Multi Agent. Looking forward, EVADE-Bench offers a rigorous foundation for evaluating and advancing multimodal moderation systems, highlighting key bottlenecks and design principles—such as clearer taxonomy and robust error handling—for building safer, more reliable AI systems.

## ETHICS STATEMENT

During the development of the EVADE-Bench, we strictly adhered to ethical guidelines and legal regulations, ensuring fairness, transparency, inclusivity, and respect for all stakeholders. The

EVADE-Bench may contain expressions and visual materials influenced by objective factors such as the time of collection, cultural context, and business scenarios. These representations and viewpoints do not reflect the value orientation of the data providers. We are committed to ongoing monitoring and refinement to mitigate such biases. Furthermore, we encourage users of the dataset to exercise responsible use and to consider the ethical implications of their work, particularly in applications that may affect individuals or communities.

## REPRODUCIBILITY STATEMENT

This work prioritizes reproducibility through comprehensive documentation of all experimental components. In the appendix, we provide detailed experimental specifications including random seed settings, API source information, and inference temperature parameters. For training procedures, we fully disclose the GPU configurations, number of epochs, learning rates, and batch sizes. Additionally, we present two rigorous pseudocode algorithms that explicitly describe our dataset construction process and experimental pipeline. In the supplementary materials, we provide representative image and text samples from EVADE-Bench, along with complete Python implementations for both data preprocessing and EVADE-Bench evaluation. Our implementation relies primarily on widely-available open-source libraries and frameworks. The detailed parameter settings and step-by-step procedures in our appendix should enable other researchers to replicate our results with minimal ambiguity. For experiments involving commercial API calls, we specify the exact API source. All reported results are computed with fixed seeds to ensure statistical reliability.

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

## A  STATEMENT ON THE USE OF LLMS

We employed Large Language Models (LLMs) as an auxiliary tool in preparing this manuscript. The LLM's function was strictly confined to assisting with two tasks: polishing the English prose and generating simple Python code snippets. All core research components—including the conceptualization, experimental design, data analysis, and the formulation of conclusions—were conducted exclusively by the human authors through collaborative discussion.

## B  EXPERIMENTAL SETUP DETAILS

All experiments were conducted with the temperature parameter consistently set to 0.8 across all LLMs and VLMs, using APIs provided by cloud service providers. For all scenarios where data needs to be randomly selected, we set the random seed to 42.

We employed Llama-Factory to perform Supervised Fine-tuning (SFT) on Qwen2.5-VL-7B and Qwen2.5-VL-32B models, utilizing $8 \times$ NVIDIA H20 GPUs for LoRA training. The hyperparameters were configured as follows: number of epochs=3, PyTorch version=2.6.0, per-device-train-batch-size=1, gradient-accumulation-steps=4, cutoff-len=10240, and initial learning-rate=5e-5.

## C  DATASET FILTERING AND EXPERT ANNOTATION

---

**Algorithm 1** Dataset Filtering Pipeline

---

1: **Input:** Raw Dataset $D$ without ground truth $GT$
2: **Output:** Filtered Dataset $D_{unlabeled}$ without ground truth $GT$
3: **Stage 1: ID Deduplication**
4:    $D_{img} \leftarrow \text{UniqueByID}(D_{images})$
5:    $D_{txt} \leftarrow \text{UniqueByID}(D_{texts})$
6: **Stage 2: Clustering & Sampling**
7: **for** $modality \in \{img, txt\}$ **do**
8:        $C \leftarrow \text{ClusterInto300}(D_{modality})$
9:        $D_{sampled} \leftarrow \emptyset$
10:    **for** each cluster $c \in C$ **do**
11:            $D_{sampled} \leftarrow D_{sampled} \cup \text{RandomSample}(c, 60)$
12:    **end for**
13:        $D_{balanced}[modality] \leftarrow D_{sampled}$
14: **end for**
15: **Stage 3: Model Validation with All Models**
16:        $Models \leftarrow \{Qwen\text{-}7B, Qwen\text{-}72B, ..., Claude\text{-}3.7\}$
17:        $D_{unlabeled} \leftarrow \emptyset$
18: **for** each sample $s \in D_{balanced}$ **do**
19:        $predictions \leftarrow \{m(s)|m \in Models\}$
20:    **if** $\text{HasDisagreement}(predictions)$ **then**
21:            $D_{unlabeled} \leftarrow D_{unlabeled} \cup \{s\}$
22:    **end if**
23: **end for**
24: **return** $D_{unlabeled}$

---

**Algorithm 2** Expert Annotation Pipeline

1: **Input:** Filtered dataset $D_{unlabeled}$ needing expert annotation
2: **Output:** Annotated dataset $D_{final}$ with ground truth $GT$
3: $D_{final} \leftarrow \emptyset$
4: **for** each sample $i$ in $D_{unlabeled}$ **do**
5:     **if** $i$ is text **then**
6:         $P^i_{model} \leftarrow \{P^i_{GPTo1}, P^i_{DeepSeekR1}, P^i_{QwenMax}\}$
7:     **else**
8:         $P^i_{model} \leftarrow \{P^i_{GPT4o}, P^i_{Claude3.7}, P^i_{Gemini2.5}\}$
9:     **end if**
10:     $A^i_{initial} \leftarrow$ Initial expert annotation
11:     **if** $A^i_{initial} = P^i_{model1} = P^i_{model2} = P^i_{model3}$ **then**
12:         $GT^i \leftarrow A^i_{initial}$
13:     **else**
14:         $A^i_{revised} \leftarrow$ Expert's revised annotation
15:         **if** $A^i_{revised} = P^i_{model1} = P^i_{model2} = P^i_{model3}$ **then**
16:             $GT^i \leftarrow A^i_{revised}$
17:         **else**
18:             $A^i_{ensemble} \leftarrow$ Five expert annotations
19:             $GT^i \leftarrow \text{mode}(A^i_{ensemble})$
20:         **end if**
21:     **end if**
22:     Add sample $i$ with its $GT^i$ to $D_{final}$
23: **end for**
24: **return** $D_{final}$

## D    FINE-GRAINED BREAKDOWN OF THE CATEGORY DISTRIBUTION

The following shows the distribution of data quantities for different ground-truth labels across the 6 subdivided data types in EVADE-Bench. Note that each sample has multi-class ground-truth labels.

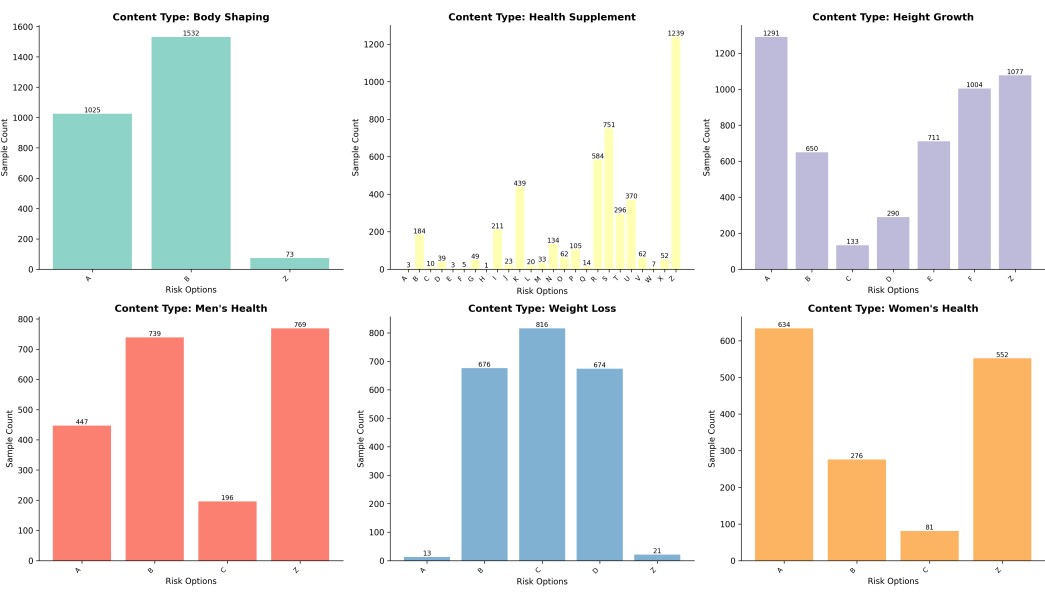

Figure 3: Fine-grained breakdown of the category distribution of the EVADE-Bench.

# E   DETAILED PERFORMANCE OF ALL MODELS ON EVADE-BENCH

In the previous main paper content, due to space limitations, we only presented the average performance of all models across six categories in EVADE-Bench. Here we will focus on showcasing the performance of each model in six categories.

Table 8: VLMs Main results on Single-Violation task. The values to the left of the slash indicate partial accuracy, while those to the right indicate full accuracy.

| | Overall | Body. | Women. | Height. | Men. | Weight. | Health. |
|---|---|---|---|---|---|---|---|
| *Open Source* | | | | | | | |
| MiniCPM-V2.6 | 44.15 / 12.37 | 44.85 / 20.15 | 52.66 / 10.42 | 58.71 / 2.13 | 43.96 / 18.81 | 55.36 / 2.58 | 26.01 / 17.54 |
| InternVL3-8B | 42.48 / 18.87 | 40.82 / 15.70 | 49.03 / 33.59 | 45.37 / 10.74 | 48.33 / 28.48 | 53.37 / 2.66 | 33.33 / 23.28 |
| InternVL3-14B | 51.20 / 23.23 | 48.59 / 19.63 | 53.67 / 39.15 | 61.92 / 22.38 | 55.06 / 27.91 | 60.85 / 2.83 | 38.54 / 24.77 |
| InternVL3-38B | 49.19 / 21.40 | 49.81 / 17.53 | 56.14 / 40.46 | 55.73 / 14.09 | 53.62 / 32.51 | 57.02 / 1.16 | 37.22 / 24.67 |
| DeepSeek-VL2 | 29.12 / 12.24 | 36.46 / 16.68 | 38.15 / 25.17 | 34.34 / 8.75 | 38.32 / 21.63 | 43.72 / 4.07 | 10.22 / 7.25 |
| Qwen2.5-VL-7B | 44.52 / 19.96 | 34.77 / 9.84 | 47.72 / 33.82 | 50.04 / 12.46 | 54.20 / 29.80 | 49.71 / 1.66 | 38.44 / **28.17** |
| Qwen2.5-VL-32B | 52.39 / 22.57 | 56.47 / 21.09 | 57.07 / 39.00 | 55.21 / 16.49 | 53.11 / 28.60 | 69.66 / 4.74 | 41.25 / 25.87 |
| Qwen2.5-VL-72B | 57.63 / 26.05 | 57.78 / 20.81 | 61.93 / 42.08 | 67.70 / **26.64** | 55.87 / 32.11 | 72.57 / 2.99 | **44.35** / 27.38 |
| *Close Source* | | | | | | | |
| GPT-4o-0806 | 58.47 / **26.96** | 70.29 / 27.79 | 61.16 / 43.32 | 64.69 / 24.16 | 55.58 / 37.11 | 75.23 / 4.57 | 42.84 / 25.97 |
| Claude-3.7 | **58.79** / 23.42 | 75.59 / 29.57 | 55.75 / 35.68 | 66.68 / 19.61 | 52.70 / 32.91 | 73.65 / 3.24 | 42.88 / 21.48 |
| Gemini-2.5-pro | 52.44 / 22.14 | **75.96** / **30.27** | 61.54 / 39.69 | **67.96** / 24.86 | 57.36 / 32.39 | **86.78** / **14.71** | 12.84 / 8.14 |
| Qwen-VL-Max | 53.38 / 25.60 | 55.62 / 29.80 | **64.17** / **49.11** | 60.52 / 19.11 | **59.15** / **39.36** | 55.20 / 1.66 | 40.08 / 22.65 |
| *Human Performance* | | | | | | | |
| Human on Image | 69.20 / 51.41 | 49.80 / 31.40 | 62.13 / 48.22 | 69.14 / 49.16 | 78.00 / 65.33 | 79.17 / 26.22 | 74.83 / 65.97 |

Table 9: LLMs Main results on Single-Violation task. The values to the left of the slash indicate partial accuracy, while those to the right indicate full accuracy.

| | Overall | Body. | Women. | Height. | Men. | Weight. | Health. |
|---|---|---|---|---|---|---|---|
| *Open Source* | | | | | | | |
| Llama3.1-8B | 35.62 / 20.93 | 64.85 / 62.38 | 68.25 / 49.29 | 33.82 / 11.03 | 37.27 / 26.38 | 38.01 / 10.86 | 17.59 / 10.61 |
| Llama3.1-70B | 38.55 / 26.23 | 51.49 / 48.51 | 49.29 / 47.39 | 25.50 / 11.03 | 46.01 / 35.12 | 52.94 / 22.85 | 27.04 / 19.92 |
| Qwen2.5-7B | 38.48 / 27.18 | 25.25 / 22.77 | 40.28 / 36.02 | 41.41 / 24.59 | 54.60 / 46.32 | 34.39 / 12.22 | 28.07 / 20.18 |
| Qwen2.5-14B | 45.39 / 28.70 | 41.58 / 38.12 | 51.18 / 41.71 | 47.92 / **29.84** | 50.61 / 37.42 | 58.37 / 21.95 | 31.18 / 18.37 |
| Qwen2.5-32B | 46.56 / 29.69 | 58.42 / 55.45 | 65.40 / 48.34 | 30.92 / 12.12 | 56.75 / 45.09 | 56.33 / 16.06 | 35.32 / **25.23** |
| Qwen2.5-72B | 49.21 / 27.85 | 59.41 / 51.49 | 65.40 / 35.07 | 41.77 / 24.41 | 57.21 / 41.87 | 61.09 / 16.06 | 33.89 / 17.08 |
| Qwen3-32B | 48.71 / 26.16 | 61.88 / 55.94 | 63.98 / 37.91 | 37.61 / 14.29 | 52.91 / 36.20 | 63.57 / 21.72 | **37.00** / 17.72 |
| Qwen3-32B_{(T)} | 47.86 / 24.71 | 60.40 / 54.46 | 69.19 / 41.71 | 36.89 / 13.56 | 51.38 / 32.98 | 63.12 / 19.91 | 34.93 / 16.04 |
| Qwen3-30B | 46.70 / 26.44 | 47.03 / 42.08 | 63.51 / 44.55 | 41.95 / 19.53 | 51.38 / 37.73 | 59.28 / 15.16 | 34.28 / 19.28 |
| Qwen3-30B_{(T)} | 47.51 / 27.64 | 40.10 / 37.13 | 63.98 / 45.02 | 43.94 / 22.42 | 52.76 / 40.49 | 59.95 / 15.84 | 35.96 / 20.05 |
| Qwen3-235B | 49.77 / 27.46 | 64.85 / 60.89 | 70.62 / 48.34 | 39.96 / 15.91 | 56.29 / 38.34 | 68.55 / 22.62 | 30.92 / 14.88 |
| Qwen3-235B_{(T)} | 50.02 / 27.50 | 62.38 / 56.93 | 70.14 / 45.97 | 41.41 / 15.55 | 56.90 / 39.88 | 65.84 / 22.62 | 32.60 / 15.65 |
| DeepSeek-V3 | 51.85 / 28.77 | 69.31 / 63.86 | 70.62 / 42.65 | **51.18** / 26.76 | 55.06 / 40.18 | 67.42 / 15.84 | 31.05 / 15.01 |
| DeepSeek-R1_{(T)} | **54.64** / 25.45 | **80.69** / **71.78** | **78.20** / 40.28 | 44.12 / 15.37 | 54.91 / 33.44 | **79.19** / 19.23 | 34.67 / 13.32 |
| *Close Source* | | | | | | | |
| GPT-o1mini | 49.28 / 29.72 | 63.86 / 56.93 | 69.67 / **65.88** | 50.63 / 28.21 | 50.92 / 34.36 | 60.63 / 20.36 | 31.05 / 15.27 |
| GPT-4.1-0414 | 52.74 / **31.59** | 61.88 / 52.48 | 72.99 / 54.98 | 43.22 / 20.98 | **58.90** / 46.32 | 71.72 / **24.66** | 35.58 / 18.89 |
| Qwen-Max | 48.29 / 31.27 | 63.86 / 58.42 | 68.25 / 54.03 | 41.59 / 25.14 | 58.59 / **47.09** | 56.56 / 16.97 | 30.14 / 17.21 |
| *Human Performance* | | | | | | | |
| Human on Text | 69.34 / 57.03 | 65.60 / 59.61 | 82.46 / 78.20 | 48.82 / 38.20 | 77.60 / 69.51 | 81.89 / 53.88 | 67.23 / 55.31 |

# F  FORMAL DEFINITION OF FOUR-LEVEL RULE SYSTEM

## F.1  BACKGROUND

To systematically analyze the multi-level reasoning that models require to handle evasive content, we formalize the complex moderation policies in EVADE-Bench into a Four-Level Rule System. This system not only defines violation types but, more critically, establishes a clear logical priority, enabling a more fine-grained evaluation of the models' reasoning processes. The core principles correspond to the evaluation tasks as follows:

- **First-Order Rule (R1)**: Direct Keyword Matching. This is the most fundamental detection level, requiring the model to identify explicit violating terms directly from text or OCR results. For instance, in the "Height Growth" category, Rule A ("Direct Height Growth Claims") is a typical R1 rule that matches terms like "grow taller" or "height increase" (see Table 10 and Table 11).
- **Second-Order Rule (R2)**: Semantic Pattern Matching. This level assesses the model's ability to understand implicit and evasive language, where the content may not contain any direct violating keywords. For example, in the same "Height Growth" category, Rule C ("Implicit Height Growth Claims") is an R2 rule. It requires the model to comprehend figurative and suggestive phrases, such as "all legs below the waist", and associate them with the intent of "height growth" (see Table 10 and Table 11).
- **Third-Order Rule (R3)**: Combinatorial Logic Judgment. This involves more advanced reasoning, requiring the model to perform a logical AND operation over multiple elements rather than relying on a single piece of information. For example, in the "Health Supplement" category, Rule I ("Stones/Nodules + Treatment Claims") is an R3 rule. Under this rule, the mere presence of "kidney stones" is not a violation; the model must also find general therapeutic claims like "remove" or "cure" to make a correct violation judgment (see Table 12 and Table 13).
- **Fourth-Order Rule (R4)**: Exemption and Override. This is the highest-priority rule, designed to simulate special cases in real-world moderation. For instance, the rules specify that if content explicitly cites authoritative sources like encyclopedias or books, it should be exempted, even if it triggers rules from the lower tiers. The model must understand this meta-rule and use it to override lower-level judgments, ultimately outputting "no violation (Z.other)".

This hierarchical system allows us to attribute each model failure to a specific capability layer. For example, a model might excel at R1 (keyword recognition) but struggle with R2 (semantic understanding), or it might handle standalone R1 and R2 rules but fail the combinatorial logic of R3. Thus, the system not only provides the theoretical foundation for EVADE-Bench but also offers a precise analytical tool for subsequent error analysis and model evaluation.

## F.2  BASIC SETS DEFINITION

Let $I$ denote the input text.

Let $C = \{A, B, C, ..., X, Z\}$ be the set of all categories. Here, each element represents a specific semantic category (e.g., $A$ = *disease*, $F$ = *fertility*, $N$ = *nutrition*, and $Z$ is reserved for "other/uncategorized" texts).

Let $K(c)$ be the set of keywords for category $c \in C$.

Let $S(c)$ be the set of semantic patterns for category $c \in C$ (e.g., templates, regular expressions, or learned semantic structures).

Let $E$ be the set of treatment effect words (e.g., "治疗(treat)", "缓解(relieve)").

Let $M$ be the set of exemption conditions (phrases that explicitly prevent classification).

## F.3  RULE DEFINITIONS

### F.3.1  FIRST-ORDER RULE (DIRECT KEYWORD MATCH)

$$[H]R_1(I, c) := \exists k \in K(c) : k \in I \tag{1}$$

This rule triggers if the input contains at least one keyword from category $c$.

### F.3.2 SECOND-ORDER RULE (SEMANTIC PATTERN MATCH)

$$[H]R_2(I,c) := \neg R_1(I,c) \wedge \exists s \in S(c) : s \text{ matches } I \tag{2}$$

Here, "$s$ matches $I$" means that the input $I$ fits a semantic pattern $s$, even if no direct keywords are found. For example, a fertility-related pattern could match "不用担心怀不上(empty nursery? not for long)" without using explicit fertility keywords.

### F.3.3 THIRD-ORDER RULE (COMBINATION RULE)

$$R_3(I,c) := \exists c_1, c_2 \in C, \exists t \in E :$$
$$(R_1(I,c_1) \vee R_2(I,c_1)) \wedge$$
$$(R_1(I,c_2) \vee R_2(I,c_2)) \wedge$$
$$t \in I \tag{3}$$

This rule is triggered when the input simultaneously involves two categories (via keyword or semantic pattern) and additionally contains a treatment effect word $t$. This captures cases where multi-category context plus explicit treatment language should lead to classification.

### F.3.4 FOURTH-ORDER RULE (EXEMPTION RULE)

$$R_4(I) := \exists m \in M : m \text{ matches } I \tag{4}$$

This rule is triggered if the input contains any exemption condition. It has the highest priority because such conditions indicate that the text should not be classified under normal categories, even if other rules would otherwise apply.

### F.4 CLASSIFICATION FUNCTION

$$F(I) = \begin{cases} \{Z\} & \text{if } R_4(I) \\ \{c \mid R_3(I,c)\} & \text{else if } \exists c : R_3(I,c) \\ \{c \mid R_1(I,c)\} & \text{else if } \exists c : R_1(I,c) \\ \{c \mid R_2(I,c)\} & \text{else if } \exists c : R_2(I,c) \\ \{Z\} & \text{otherwise} \end{cases} \tag{5}$$

Here, $Z$ explicitly denotes the "other/uncategorized" class.

### F.5 PRIORITY RELATIONSHIP

$$R_4 > R_3 > R_1 > R_2 \tag{6}$$

The exemption rule overrides all others, followed by the combination rule, then direct keywords, and finally semantic pattern matches.

### F.6 EXAMPLE

For text "不用担心怀不上(empty nursery? not for long) ":

- $R_1$ returns $\emptyset$ (no direct keywords)
- $R_2$ returns $\{F\}$ (semantic pattern matches fertility context)
- $R_3$ returns $\emptyset$ (no combination rule triggered)
- $R_4$ returns false (no exemption condition met)
- Therefore, $F(I) = \{F\}$

## G RULE PROMPT EXAMPLES

The following are the audit rules for Height Growth and Weight Loss and Health Supplement products in Single violation, where option 'Z' represents non-violation cases (or irrelevant content), and all other options represent different types of violations that need to be controlled. For more review rule prompts, please refer to our Huggingface dataset.

Table 10: The Chinese prompt of Height Growth and Weight Loss products in Single violation.

| Data Type | Chinese Prompt |
| --- | --- |
| Height Growth | # 任务描述
请你根据给定信息和关键词，仔细判断对应的管控类型。给出最终的选项（一个到多个不等）。都不相关返回Z.其他。

# 管控类型
A. 增高功效直接描述：明确使用以下词汇描述长高发育：增高、长高、长个、蹭蹭长/涨、xx岁还能长、（身高）用了高/吃了高/喝了高/贴了高、吃出/喝出/补出/贴出高个子、描述商品使用前和使用后身高差距。注意：若涉及未成年人进行此类表达，则不管控，其中未成年人相关描述包括但不限于：孩子、儿童、宝宝、青少年、发育、青春期、同龄、成长缓慢、小于等于18岁、出现未成年人的形象、身穿校服等。
B. 个子矮小描述：包括但不限于：矮个子专用、适合矮个子、告别矮小、对矮小说不/say no/说拜拜/say goodbye、不做矮个子/矮XX、不再矮、X年不长、身高不够、矮小、身高+数字、不足+数字、数字+以下等间接表达身高矮小。
C. 增高隐晦描述：包括但不限于使用以下词汇来暗示身高增长：腰以下全是腿、吃出大长腿、个头/身高不掉队。
D. 磁石矫形鞋垫：包括但不限于使用以下词汇来描述身高增高、腿型矫正：鞋垫+磁石/磁疗/火疗；鞋垫+矫正/矫形/纠正/X型腿/O型腿/罗圈腿/腿直了；带有黑色磁石的鞋垫。
E. 反科学增高增高：使用如下违反科学原理的方式来增长身高：打开骨闭合/骨骼闭合、突破遗传、二次发育/破高。
F. 增高精准功效承诺：使用数字/时间/百分数来承诺使用商品后能达到确定的增高效果(例如一喝长高、一粒长高、不管多矮，只用3支、一周长高、100%长高、睡前涂，睡醒长)。
Z. 其他：当给定信息与上述选项均无明显关联时选择此项。 |
| Weight Loss | # 任务描述
请你根据给定信息和关键词，仔细判断对应的管控类型。给出最终的选项（一个到多个不等）。都不相关返回Z.其他。

# 管控类型
A. 特定减肥商品：包括以下商品：星空咖啡、韩国星空咖啡、泰国果蔬slim7day局部减肥药、slim7day、无限畅、纤so、纤姿、司美格鲁肽、司美、口服司美、利拉鲁肽、索马鲁肽、格鲁肽、glp-1。
B. 减肥夸大描述：内容使用极端用词来承诺商品使用后的减肥效果，包括但不限于"骨瘦如柴"、"皮包骨"、"狂吃不胖/涨"、"脂肪收割机/炸弹"、"急吃/喝急瘦"、"终身/一辈子/一生不长肉"等夸大描述。使用精准数字或者时间来承诺商品使用后的减肥效果，包括但不限于"x天瘦x斤(J)/kg"、"瘦x斤(J)/kg"、"xx斤(J)/kg保 底"、"排x斤(J)/kg+油/汗/便"、"挑战x/x斤(J)/kg"等量化承诺。内容将减肥前的状态（如"体重超标"、"体重难下/不下"、"肚子大"、"不掉称"）与具体数字和量词直接关联，以误导消费者对减肥效果的预期。
C. 减肥直接描述：内容直接描述身体体重、体脂的下降、身体由胖到瘦描述，包括但不限于使用特定动词（抽、清、抗、吸、溶、排、去/祛、减、分解、燃烧/燃、绝、甩等）、效果形容词（没了、小了、细了、不见了、下降/降了、少了、掉了/掉秤、轻了、不反弹/回弹、管用/有用等）和"脂/脂肪/油/肉/肥/胖/重"结合，来直接描述减肥效果。或者直接描述身体部位（如肩部、颈部、背部、手臂/胳膊、腹部/肚子/肚肚/肚腩、腰部、大腿、小腿、肉肉）变瘦。注意：若明确与减肥无关则不管控，例如"减脂期"。
D. 减肥隐晦描述：内容暗示产品或服务可以替代传统减肥方法（如节食、运动、手术、左旋肉碱等其他减肥产品）；内容声称能够提供饱腹感、抑制食欲或控制饮食的描述；特殊减肥效果暗示描述，例如：不要溜肩、天鹅臂、蝴蝶肩、小蛮腰/腰精/a4腰、圆腰克星、腰细背薄、大象腿、裤子往下滑、摆脱厚臂、远离肩厚、厚臂不见了、肩不厚了、远离拜拜肉、不要蝴蝶袖、裤子松了、换裤子/裤头、理想体重、消腩、缩腹、soso好身材、平小腹、骨感美、摆脱大码、身材自信/好身材/好身姿、拯救身材、胖子逆袭、体重不过百、体重两位数、体重重回xx、燃烧卡路里、胖子有救了。
Z. 其他：当给定信息与上述选项均无明显关联时选择此项。 |

Table 11: The English prompt of Height Growth and Weight Loss products in Single violation.

| Data Type | English Prompt |
|---|---|
| Height Growth | # Task Description
Please carefully evaluate the control type based on the given information and keywords. Provide the final option(s) (ranging from one to multiple). Select Z.Others if none are relevant.

# Violation Type
A. Direct Height Growth Claims: Explicit use of terms describing height increase such as "grow taller," "height increase," "growth spurt," "can still grow at XX age," "gained height from using/eating/drinking/applying," "achieve tallness through consumption/supplements/applications," or descriptions comparing height before and after product use. Note: These restrictions do not apply to expressions involving minors, including references to children, youth, babies, adolescents, development, puberty, peer groups, slow growth, ages ⩽18, images of minors, or school uniforms.
B. Short Stature References: Including but not limited to: "for short people," "suitable for short stature," "goodbye to being short," "say no/goodbye to shortness," "no more being short," "X years without growth," "insufficient height," "short stature," "height + number," "below + number," or similar indirect expressions about short stature.
C. Implicit Height Growth Claims: Including but not limited to suggestive phrases like "legs for days," "achieve long legs through consumption," "keep up in height/stature."
D. Magnetic Orthopedic Insoles: Including but not limited to descriptions of height increase or leg correction using terms like: "insoles + magnetic/magnetotherapy/heat therapy;" "insoles + correction/orthopedic/straightening/X-legs/O-legs/bowlegs/straight legs;" insoles with black magnets.
E. Unscientific Growth Claims: Using scientifically unsound methods for height increase such as "opening bone closure," "overcoming genetics," "second growth phase/breakthrough."
F. Specific Height Growth Promises: Using numbers/time/percentages to promise definite height increase effects (e.g., "grow taller with one drink," "just 3 doses regardless of height," "grow in one week," "100% growth guarantee," "apply before sleep, grow overnight").
Z. Others: Select this option when the given information shows no clear connection to any of the above categories. |
| Weight Loss | # Task Description
Please carefully evaluate the control type based on the given information and keywords. Provide the final option(s) (ranging from one to multiple). Select Z.Others if none are relevant.

# Violation Type
A. Specific Weight Loss Products: Including: Space Coffee, Korean Space Coffee, Thailand Slim7day local weight loss medicine, Slim7day, Unlimited Flow, Slimso, Xianzhi, Semaglutide, Sema, oral Sema, Liraglutide, Somaglutide, Glutide, GLP-1.
B. Exaggerated Weight Loss Claims: Content using extreme terminology to promise weight loss results, including but not limited to "skin and bones," "fat-burning machine/bomb," "eat without gaining weight," "lifetime/permanent slimness," etc. Claims using specific numbers or timeframes, such as "x pounds/kg in x days," "guaranteed x pounds/kg loss," "challenge x pounds/kg," etc. Content that directly correlates initial weight conditions (e.g., "overweight," "stubborn weight," "big belly") with specific numbers to mislead consumer expectations.
C. Direct Weight Loss Descriptions: Content directly describing weight or body fat reduction, including specific verbs (extract, clear, resist, absorb, dissolve, eliminate, reduce, break down, burn, etc.) combined with effect adjectives (gone, smaller, thinner, decreased, dropped, lighter, no rebound, effective, etc.) and terms like "fat/flesh/weight." Also includes direct descriptions of body part reduction (shoulders, neck, back, arms, abdomen/belly, waist, thighs, calves). Note: Excluded if clearly unrelated to weight loss, e.g., "fat-burning period."
D. Implicit Weight Loss Claims: Content suggesting products or services can replace traditional weight loss methods (diet, exercise, surgery, L-carnitine, etc.); claims about providing satiety, suppressing appetite, or controlling diet; implicit descriptions of special weight loss effects, such as: "swan arms," "butterfly shoulders," "tiny waist," "elephant legs," "loose pants," "ideal weight," "belly reduction," "bone-thin beauty," "body confidence," "weight under 100," "calorie burning," etc.
Z. Others: Select when given information shows no clear connection to above categories. |

Table 12: The Chinese prompt of Health products in Single violation.

| Data Type | Chinese Prompt |
| --- | --- |
| Health Supplement | # 任务描述
请你根据给定信息和关键词，仔细判断对应的管控类型。给出最终的选项（一个到多个不等）。都不相关返回Z.其他。

# 管控类型
- A.甲乙丙类传染病
- B.心脏类疾病：涉及心脏相关的疾病问题（包含心梗、心绞、冠心、心率、心力、血栓、心慌、心悸、胸闷等），同时还包含血管硬化、血管堵塞、血粘稠等血管血液的疾病问题和因此衍生的各种治疗用语与治疗手段。
- C.严重神经系统疾病：仅涉及神经系统相关的脑卒中、老年痴呆、中风、脑膜炎、脑梗、脑萎缩这几种严重疾病和因此衍生的各种治疗用语与治疗手段。
- D.严重内分泌疾病：仅涉及糖尿病酮症酸中毒、甲亢、甲减、桥本甲状腺等严重疾病和因此衍生的各种治疗用语与治疗手段。
- E.严重创伤和烧伤：仅涉及多发性创伤、大面积烧伤和因此衍生的各种治疗用语与治疗手段。
- F.生育类疾病：仅涉及生育的疾病问题（如不孕、不育、多囊卵巢综合征、精子畸形等）和因此衍生的各种治疗用语与治疗手段。
- G.癌症肿瘤类疾病：涉及人体各个部位的瘤、肿瘤、癌症、癌变、癌扩散等疾病和因此衍生的各种治疗用语与治疗手段。
- H.严重精神类疾病：仅涉及重度抑郁症、精神分裂等严重疾病和因此衍生的各种治疗用语与治疗手段。
- I.结石结节类疾病+通用治疗效果：涉及人体结石或结节类疾病问题的同时，还需要出现通用治疗效果词。比如仅描述"肾结石"不属于该分类，但既描述"肾结石"又提及"清除"的字眼，就属于该分类。
- J.痔疮类疾病+通用治疗效果：涉及人体痔疮类疾病问题的同时，还需要出现通用治疗效果词。比如仅描述"小肉球"不属于该分类，但既描述"小肉球"又提及"根治"的字眼，就属于该分类。
- K.生殖器官疾病+通用治疗效果：涉及男性生殖器官疾病或女性生殖器官疾病的同时，还需要出现通用治疗效果词。比如仅描述"尿道"不属于该分类，但既描述"尿道"又提及"突破"的字眼，就属于该分类。它与上面生育类疾病的区别在于，生殖器官类疾病不包含生育问题。
- L.幽门螺旋杆菌+通用治疗效果：涉及幽门螺旋杆菌类疾病的同时，还需要出现通用治疗效果词。比如仅描述"幽"不属于该分类，但既描述"幽"又提及"清"或"去"等字眼，就属于该分类。
- M.五官类疾病：涉及牙齿、耳朵、鼻子、头发、眼睛等人体五官的疾病问题和因此衍生的各种治疗用语与治疗手段。
- N.三高类疾病：涉及血压、血脂、血糖等疾病问题和因此衍生的各种治疗用语与治疗手段。
- O.肺部呼吸类疾病：涉及呼吸窘迫综合症ARDS、肺炎、肺病COPD、以及所有人体肺部器官的疾病问题和因此衍生的各种治疗用语与治疗手段。
- P.胃部消化类疾病：涉及急性胰腺炎、消化道出血、以及所有人体胃部器官的疾病问题和因此衍生的各种治疗用语与治疗手段。
- Q.肝部类疾病：涉及肝衰竭、肝硬化、以及所有肝部器官的疾病问题和因此衍生的各种治疗用语与治疗手段。
- R.关节类疾病：涉及关节疼痛、增生、麻木、畸形、肿痛、尿酸等体现在关节上的疾病和因此衍生的各种治疗用语与治疗手段。
- S.手脚皮肤类疾病：涉及指甲、灰甲、空甲、脚气、手脚糜烂、水泡、藓类、疣疮类、湿疹类、抑菌止痒类、狐臭类疾病问题和因此衍生的各种治疗用语与治疗手段。
- T.轻度疾病：涉及感冒、鼻炎、焦虑症、失眠、睡不着、深睡等一系列睡眠疾病问题和因此衍生的各种治疗用语与治疗手段。
- U.医用级描述：包含"医用级"、"医疗级"、"治愈"、"疗效"、"神药"、"奇效"、"专用药"等含义的内容，比如"瘙痒特效药"、"专攻哮喘"、"血糖百试百灵"等宣传描述。
- V.代替医疗类：包含"取代"、"比*好用"、"告别"等表达，并提及以下药品名称(他汀、双胍、4联/四联、甲钴胺、地屈孕酮、非布司他、二胍、黄体酮、所有的药和所有的疫苗)或医疗手段(手术、开刀、打针、注射、拉皮、拍CT、戴助听、做耳蜗、复查、胃镜、医美、所有的针（如水光针、美白针）)，比如仅描述"疫苗"不属于该分类，但既描述"疫苗"又提及"吊打"的字眼，就属于该分类。
- W.代替医美类：包含"填充"、"膨胀"、"丰满"、"挺起"、"缩小"等形状变化的表达，同时需要提及以下身体部位(胸部、鼻子、眼睛)，比如仅描述"胸"不属于该分类，但既描述"挺"又提及"胸"的字眼，就属于该分类。
- X.无副作用类描述：包含"无副"、"无副作用"、"无风险"、"零风险"等表达商品无副作用和风险的内容，或者包含"无药"、"无药物残留"、"无毒"、"0化学"、"0农药"等表达商品无化学或农药添加剂的内容，或者包含"安全"、"安全性高"、"副作用小"等表达商品使用风险低的内容。
- Z. 其他：当给定信息与上述选项均无明显关联时选择此项。

# 注意事项
1. 谨慎判断:若无法正确识别，可能会加重患者病情或导致其他严重后果，请务必谨慎、细致地判断。
2. 权威引用:若文本明确引用百度百科、搜狗百科、维基百科、书籍内容或书名等权威信息来源，判断为"不相关"返回Z.其他。
3. 科普内容:若文本仅为科普（如书籍、电影等），且未出现具体商品名称，判断为"不相关"返回Z.其他。
4. 患病人群描述:若文本仅描述患病人群（如描述艾滋病人、心脏病患者的书籍或活动），判断为"不相关"返回Z.其他。
5. 通用治疗效果:文本可能会出现通用治疗效果相关的词汇，这些效果表现类词汇主要用于描述事物在功能或状态上的变化，突出其带来的积极效果。它们可以分为多个类别:上升类词汇如"拯救"、"改善"、"提高"、"增强"、"加强"、"提升"等，强调对某种能力或状态的提升和优化；下降类词汇如"抗"、"抑制"、"降"、"镇"、"缓解"、"控制"等，侧重于减少、不良因素的抑制或控制；消除类词汇如"告别"、"根治"、"消失"、"彻底"、"治愈"、"摆脱"、"解决"等，强调对问题或障碍的彻底消除和根除；用量类词汇如"仅需"、"只要"、"只需"等，突出使用的简便性和低成本。此外，其他词汇如"有效率"、"治愈率"等，强调效果的高效性和成功率。这些词汇的共同特征在于通过具体的动词和形容词，传达出显著的效果和优势。 |

Table 13: The English prompt of Health products in Single violation.

| Data Type | English Prompt |
|---|---|
| Health Supplement | # Task Description
Please carefully evaluate the control type based on the given information and keywords. Provide the final option(s) (ranging from one to multiple). Select Z.Others if none are relevant.

# Violation Type
A. Class A/B/C Infectious Diseases
B. Cardiac Diseases: Heart-related conditions (including myocardial infarction, angina, coronary disease, heart rate/rhythm issues, heart failure, thrombosis, palpitations, chest tightness), plus vascular diseases like arteriosclerosis, blockage, and blood viscosity issues, including related treatments.
C. Severe Neurological Disorders: Limited to stroke, dementia, cerebral meningitis, cerebral infarction, brain atrophy, and associated treatments.
D. Severe Endocrine Disorders: Limited to diabetic ketoacidosis, hyperthyroidism, hypothyroidism, Hashimoto's thyroiditis, and associated treatments.
E. Severe Trauma and Burns: Limited to multiple trauma injuries, extensive burns, and associated treatments.
F. Reproductive Disorders: Limited to fertility issues (such as infertility, PCOS, sperm abnormalities) and associated treatments.
G. Cancer and Tumors: All types of tumors, cancers, malignant transformations, metastasis, and associated treatments.
H. Severe Mental Disorders: Limited to major depression, schizophrenia, and associated treatments.
I. Stones/Nodules + Treatment Claims: References to body stones or nodules combined with general treatment effect terms. Example: "kidney stones" alone doesn't qualify, but "removing kidney stones" does.
J. Hemorrhoids + Treatment Claims: References to hemorrhoids combined with general treatment effect terms. Example: "hemorrhoids" alone doesn't qualify, but "cure hemorrhoids" does.
K. Genital Diseases + Treatment Claims: References to male or female genital conditions (excluding fertility issues) combined with general treatment effect terms. Example: "urethral" alone doesn't qualify, but "breakthrough urethral treatment" does.
L. H. Pylori + Treatment Claims: References to H. pylori combined with general treatment effect terms. Example: "H. pylori" alone doesn't qualify, but "eliminate H. pylori" does.
M. Sensory Organ Diseases: Conditions affecting teeth, ears, nose, hair, eyes, and associated treatments.
N. "Three Highs" Diseases: Blood pressure, blood lipids, blood sugar disorders and treatments.
O. Pulmonary/Respiratory Diseases: ARDS, pneumonia, COPD, and other lung conditions and treatments.
P. Gastrointestinal Diseases: Acute pancreatitis, digestive tract bleeding, and other gastric conditions and treatments.
Q. Liver Diseases: Liver failure, cirrhosis, and other hepatic conditions and treatments.
R. Joint Diseases: Joint pain, hyperplasia, numbness, deformity, inflammation, uric acid issues, and treatments.
S. Hand/Foot/Skin Conditions: Nail conditions, athlete's foot, skin ulcers, blisters, fungal infections, warts, eczema, antibacterial treatments, body odor issues.
T. Minor Ailments: Common cold, rhinitis, anxiety, insomnia, sleep disorders, and treatments.
U. Medical-Grade Claims: Terms like "medical-grade," "therapeutic," "miracle cure," "specialized medicine," etc.
V. Medical Alternative Claims: Claims about replacing conventional treatments/medications (statins, metformin, vaccines, surgeries, injections, etc.) using terms like "replace," "better than," "goodbye to."
W. Cosmetic Alternative Claims: Body modification claims (chest, nose, eyes) using terms like "fill," "enlarge," "firm up."
X. No Side Effects Claims: Claims of "no side effects," "risk-free," "chemical-free," "pesticide-free," "safe," "high safety."

# Important Guidelines
1. Careful Assessment: Exercise caution in identification as incorrect judgment may worsen patient conditions or lead to serious consequences. Thorough and detailed evaluation is essential.
2. Authoritative Citations: Content explicitly citing authoritative sources (e.g., Baidu Encyclopedia, Sogou Encyclopedia, Wikipedia, books) should be classified as "unrelated" (return Z.Others).
3. Educational Content: Pure educational materials (e.g., books, films) without specific product mentions should be classified as "unrelated" (return Z.Others).
4. Patient Population Descriptions: Content merely describing patient populations (e.g., books or activities about HIV or heart disease patients) should be classified as "unrelated" (return Z.Others).
5. General Treatment Effect Terms: Content may include words describing therapeutic effects, which can be categorized as follows: Improvement terms: "rescue," "enhance," "improve," "strengthen," "boost," "elevate" Reduction terms: "resist," "inhibit," "decrease," "calm," "relieve," "control" Elimination terms: "goodbye to," "cure," "disappear," "complete," "heal," "overcome," "solve" Dosage terms: "just need," "only requires," "simply needs" Efficacy terms: "effectiveness rate," "cure rate" These terms typically use specific verbs and adjectives to communicate significant benefits and advantages. |

## H   ALL MODELS IN EXPERIMENTS AND SOME CASES OF EVADE-BENCH

Table 14: Baseline Model List.

| LLMs | VLMs |
| --- | --- |
| Qwen-2.5-7B (Qwen et al., 2025) | MiniCPM-V2.6-8B (Yao et al., 2024) |
| Qwen-2.5-14B (Qwen et al., 2025) | InternVL3-8B (Zhu et al., 2025) |
| Qwen-2.5-32B (Qwen et al., 2025) | InternVL3-14B (Zhu et al., 2025) |
| Qwen-2.5-72B (Qwen et al., 2025) | InternVL3-38B (Zhu et al., 2025) |
| Qwen-3-32B (Yang et al., 2025) | DeepSeek-VL2-27B (Wu et al., 2024) |
| Qwen-3-30B-A3B (Yang et al., 2025) | Qwen-2.5-VL-7B (Bai et al., 2025a) |
| Qwen-3-235B-A22B (Yang et al., 2025) | Qwen-2.5-VL-32B (Bai et al., 2025a) |
| DeepSeek-V3-671B (DeepSeek-AI et al., 2025b) | Qwen2.5-VL-72B (Bai et al., 2025a) |
| Llama-3.1-8B (Grattafiori et al., 2024) | GPT-4o-0806 (OpenAI et al., 2024a) |
| Llama-3.1-70B (Grattafiori et al., 2024) | Claude-3.7-sonnet (claude37, 2025) |
| DeepSeek-R1-671B (DeepSeek-AI et al., 2025a) | Gemini-2.5-Pro (Comanici et al., 2025) |
| GPT-o1mini-0912 (openai, 2024) | Qwen-VL-Max |
| GPT-4.1-0414 (openai, 2025) | |
| Qwen-Max | |

**Height Growth Product**

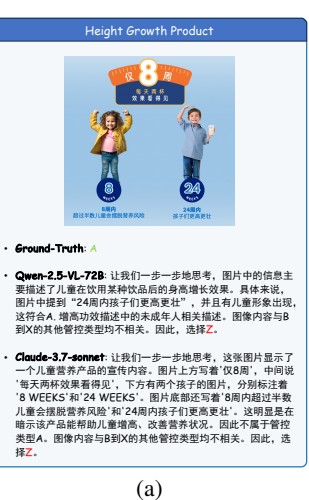

- **Ground-Truth**: A
- **Qwen-2.5-VL-72B**: 让我们一步一步地思考，图片中的信息主要描述了儿童在饮用某种饮品后的身高增长效果。具体来说，图片中提到"24周内孩子更高更壮"，并且有儿童形象出现，这符合于A.高功效描述中的未成年人相关描述。图像内容与B到X的其他管控类型均不相关。因此，选择Z。
- **Claude-3.7-sonnet**: 让我们一步一步地思考，这张图片显示了一个儿童营养产品的宣传内容。图片上方写着"从8周"，中间说"每天两杯效果看得见"，下方有两个大的图片，分别标注着"8 WEEKS"和"24 WEEKS"。图片底部还写着"8周内超过半数儿童会摆脱营养风险"和"24周内孩子们更高更壮"。这明显是在暗示该产品能帮助儿童增高、改善营养状况。因此不属于管控类型A。图像内容与B到X的其他管控类型均不相关。因此，选择Z。

(a)

**Height Growth Product**

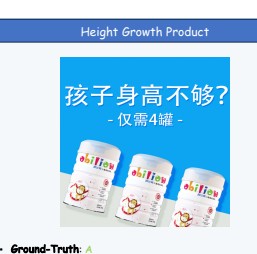

- **Ground-Truth**: A
- **Qwen-2.5-VL-72B**: 让我们一步一步地思考，图片中的文字信息主要描述了一款为"欧拉熊儿童成长奶粉"的产品，针对孩子身高不够的问题，提出"仅需4罐"的解决方案。首先，这明确指向了儿童群体，因此不涉及成年人的增高问题，排除了A选项中关于成年人身高的管控。其次，该产品并未涉及丰胸、减肥、壮阳、细阴等功效描述，因此选择Z选项。
- **Claude-3.7-sonnet**: 让我们一步一步地思考，查看这张图片和文字内容。图片展示了一款为'obillion'的儿童产品，文字标题为'孩子身高不够？'，副标题为'仅需4罐'。这明显是在暗示使用该产品可以增加儿童身高。根据管控类型说明，A类（增高功效描述）提到'若涉及未孩子/儿童/少年/青春期等未成年人进行此类表达，则不管控'，则选择Z选项。

(b)

**Weight Loss Product**

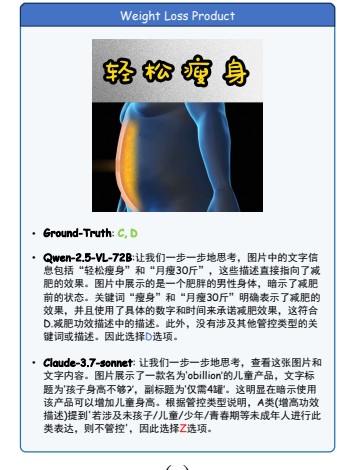

- **Ground-Truth**: C, D
- **Qwen-2.5-VL-72B**: 让我们一步一步地思考，图片中的文字信息包括"轻松瘦身"和"月瘦30斤"，这些描述直接指向了减肥的效果。图片中展示的是一个肥胖的男性身体，暗示了减肥前的状态。关键词"瘦身"和"月瘦30斤"明确表示了减肥的效果，并且使用了具体的数字和时间来承诺减肥效果，这符合D.减肥功效描述中的描述。此外，没有涉及其他管控类型的关键词或描述。因此选择D选项。
- **Claude-3.7-sonnet**: 让我们一步一步地思考，查看这张图片和文字内容。图片展示了一款名为'obillion'的儿童产品，文字标题为'孩子身高不够？'，副标题为'仅需4罐'。这明显是在暗示使用该产品可以增加儿童身高。根据管控类型说明，A类（增高功效描述）提到'若涉及未孩子/儿童/少年/青春期等未成年人进行此类表达，则不管控'，因此选择Z选项。

(c)

**Health Supplement Product**

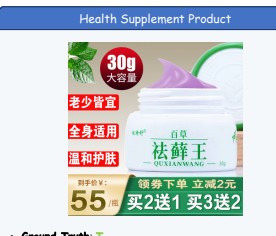

- **Ground-Truth**: T
- **Qwen-2.5-VL-72B**: 让我们一步一步地思考，图片中的产品名为"祛藓王"，主要描述其容量为30g，适合老少皆宜、全身适用，并强调温和护肤。从产品名称和描述来看，该产品主要针对皮肤问题，特别是与藓类相关的皮肤问题。没有涉及任何与增高、丰胸、减肥、壮阳、细阴等功效的描述，也没有涉及任何严重疾病或医疗级描述。因此，该产品与上述管控类型中的A到V选项均无明显关联。因此选择Z选项。
- **Claude-3.7-sonnet**: 让我们一步一步地思考，首先观察图片内容。这是一张护肤品广告图，主要展示了一款叫"祛藓王"的产品。图片中标注了以下信息："30g大容量"、"老少皆宜"、"全身适用"、"温和护肤"，还有价格信息'到手价¥55/瓶'，以及促销信息'买2送1 买3送2'。需要判断这些信息是否涉及管控类型。从产品名称'祛藓王'和宣传来看，可能暗示有治疗某种皮肤问题的功效。因此选T选项。

(d)

**Body Shaping Products**

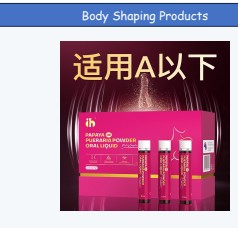

- **Ground-Truth**: B
- **Gpt-4o**: 让我们一步一步地思考，图片中的产品以木瓜和葛根为主要成分，并以粉剂口服液形式呈现。包装上写明'适用A以下'，暗示适用于身高在'A'以下的人群，这种表达可能涉及对身高的间接描述。此外，该产品可能也与丰胸有关，因为木瓜和葛根通常与丰胸功效相关。综合考虑这些因素，可能涉及增高和丰胸功效的描述。因此，选择A、B。
- **Qwen2.5-VL-7B**: 图片展示了一款名为'PAPAYA AND PUERARIA POWDER ORAL LIQUID'的产品，包装上没有提及任何与身高、胸部、减肥、壮阳、细阴、疾病治疗等相关的描述。产品描述中也没有涉及任何医疗级、治愈、疗效字样，也没有提及任何副作用或风险。因此，该产品不属于任何特定的管控类型。因此，选择Z。

(e)

**Men's Health Products**

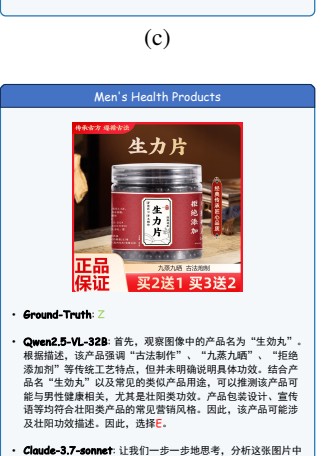
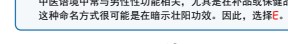

- **Ground-Truth**: Z
- **Qwen2.5-VL-32B**: 首先，观察图像中的产品名为"生劲丸"。根据描述，该产品强调"古法制作"、"九蒸九晒"、"拒绝添加剂"等传统工艺特点，但并未明确说明具体功效。结合产品名"生劲丸"以及常见的类似产品用途，可以推测该产品可能与男性健康相关，尤其是壮阳类功效。产品包装设计、宣传语等均符合壮阳类产品的常见营销风格。因此，该产品可能涉及壮阳功效的描述。因此，选择E。
- **Claude-3.7-sonnet**: 让我们一步一步地思考，分析这张图片中的产品信息和描述。标签上有'传承古方 道循古法'、'经典承匠心品质'等字样。还有"拒绝添加"、"九蒸九晒 古法炮制"等说明。"生力片"暗示了增强体力或精力的功效。"生力"一词在中医语境中常与男性性功能相关，尤其是在补品或保健品中。这种命名方式很可能是在暗示壮阳功效。因此，选择E。

(f)

Figure 4: Six Cases from All-in-One task on EVADE-Bench.

