# OpenReview forum: "EVADE-Bench: Multimodal Benchmark for Evasive Content Detection in E-Commerce Applications"
_ICLR.cc/2026/Conference — Submitted to ICLR 2026_

### Official Review · Reviewer_9eFs · 2025-10-24

**Soundness:** 3
**Presentation:** 2
**Contribution:** 2
**Rating:** 2
**Confidence:** 4

**Summary:**

This paper presents EVADE-Bench, a Chinese multimodal benchmark for detecting evasive content in e-commerce advertisements, where sellers deliberately obscure prohibited claims.
The dataset contains 2.8K text and 13.9K image samples annotated by domain experts under six product categories.
Two evaluation settings are designed: Single-Violation and All-in-One.
The authors benchmark 26 LLMs and VLMs, analyze their accuracy under Partial/Full match metrics, and further explore RAG/SFT enhancements.

**Strengths:**

(1) EVADE-Bench tackles a practically important and underexplored safety problem (e-commerce content moderation) and provides the first large-scale Chinese multimodal dataset aligned with legal advertising rules.

(2) The split between Single-Violation and All-in-One tasks provides interesting insights into how rule clarity and context length affect model reasoning.

(3) The discussion on hallucination, omission, OCR distortion, and rule-matching bias provides useful diagnostics for practitioners.

**Weaknesses:**

1. The work primarily repackages an individual content-moderation pipeline. It lacks a methodological or conceptual contribution to multimodal reasoning or safety. The dataset is heavily tied to Chinese e-commerce advertising and cannot generalize to other safety or reasoning contexts. Thus, it does not qualify as a "foundation benchmark", but rather a domain-specific dataset.

2. The results are broad but shallow, no causal or ablation analysis of why model fail, or what architecture factors matter. The paper's Section 5.5 presents a discussion of adopting a two stage VLM --> LLM pipeline inspired by Prism. However, no corresponding experiments or quantitative results are reported. The discussion remains conceptual, which weakens the empirical rigor of the paper's analytical claims.

3. The paper does not disclose the exact data source (which platforms, how samples were licensed, whether commercial content was reproduced), raising reproducibility and ethical concerns.

**Questions:**

1. Could the authors clarify the data collection sources and the legal/ethical permissions involved?

2. Have the authors tried multimodal decomposition (VLM→LLM reasoning) empirically, or is this purely conceptual?

---

> ### Author Response · Authors · 2025-11-21
> **Response to Reviewer 9eFs**
>
> We thank the reviewer for the critical feedback. We understand your concerns regarding the benchmark's scope and depth. However, we respectfully disagree that this is merely a repackaged moderation dataset. Below, we provide new experimental evidence (addressing the decomposition pipeline) and clarify the methodological contributions to demonstrate that EVADE-Bench serves as a rigorous probe for generalizable multimodal reasoning under adversarial ambiguity.
>
> ***
>
> **W2 & Q2: Empirical validation of multimodal decomposition (VLM to LLM)**
>
> To verify the effectiveness of the multimodal decoupling approach (Section 5.5), we conducted experiments using GPT-4o and Claude-3.7 specifically for converting images into pure text descriptions, and then employed different powerful LLMs to perform multi-class inference on the generated text alone, without access to the images themselves. **Please refer to "Experimental Validation of Two-Stage VLM Decomposition" in the global official comment.**
>
> ***
>
> **W1: Methodological contribution and Foundation Benchmark qualification**
>
> We respectfully argue that EVADE-Bench is not just a domain-specific dataset, but a foundational testbed for Adversarial Information Restoration, a critical gap in current multimodal safety evaluation.
>
> Methodological Gap: Existing benchmarks (e.g., MMMU, MathVista) test reasoning in clear contexts. Safety benchmarks (e.g., SafetyBench) test robustness against overt harm (violence, hate). EVADE-Bench targets a distinct blind spot: Deception Detection. **It requires the model to first restore ground truth from deliberately obfuscated inputs (e.g., visual puns, homophones, occlusions) before applying logic. We define this capability as Restoration-then-Reasoning. This capability is fundamental to any robust foundation model, not just for e-commerce, but for disinformation detection, copyright protection, and anti-fraud systems.**
>
> Generalizability: **While sourced from e-commerce, the core challenge of grounding abstract visual cues into complex logical rules is domain-agnostic**. Just as medical benchmarks probe biological knowledge, EVADE-Bench probes the model's resistance to semantic noise. We are developing an English version but prioritized Chinese because it presents a Hard Mode for ECD due to its logographic nature. In the 2D structure of Chinese characters, evasive operations like complex character splitting and visual puns appear much more frequently than in alphabetic languages. Models that succeed on the semantic density of Chinese EVADE-Bench are likely to possess robust reasoning capabilities transferrable to other languages.
>
> Beyond the dataset construction, our work offers some insights and contributions. First, we discovered the Strong-Model Bias, a counter-intuitive phenomenon where stronger models like GPT-4o perform worse on prompt adherence than weaker models in specific scenarios **(lines 432-441)**. Second, we propose a novel Human-Model Collaborative Pipeline **(Algorithms 1 & 2 in Appendix C)**. We observed that while models possess text reasoning capabilities comparable to humans, humans have far superior multimodal understanding. By combining these strengths, we ensured the high quality of EVADE-Bench annotations, achieving a Krippendorff's Alpha greater than 0.71. Third, we found that category overlap in multi-class classification tasks poses a greater challenge to large models than long-context understanding **(Section 5.2)**. Fourth, we propose the insight of decoupling multimodal reasoning. Since the core challenge of ECD lies in the semantic ambiguity of the input, we suggest separating the steps of seeing (image description) and reasoning (logic application) to enhance model capabilities, rather than relying on single-step integrated inference **(Section 5.5)**.
>
> ***
>
> **W3 & Q1: Data source transparency and ethics**
>
> We strictly adhere to ethical guidelines and reproducibility standards.
>
> **Source Nature**: All samples are publicly accessible commercial product listings. No private or user-generated content (UGC) containing Personal Identifiable Information (PII) is included.
>
> **Permissions**: We have obtained approval from our institution's committee. The data collection complies with the Fair Use principles for academic research.
>
> **Release Plan**: Upon acceptance, we will release the dataset under a CC-BY-NC 4.0 license with a detailed datasheet, disclosing the platform categories (anonymized) to ensure reproducibility while respecting commercial sensitivities.

---

### Official Review · Reviewer_kXw3 · 2025-10-25

**Soundness:** 2
**Presentation:** 2
**Contribution:** 2
**Rating:** 4
**Confidence:** 4

**Summary:**

This paper introduces EVADE-Bench, the first expert-curated Chinese multimodal benchmark designed to evaluate the performance of Large Language Models (LLMs) and Vision-Language Models (VLMs) in detecting evasive content in e-commerce scenarios. The benchmark comprises 2,833 text samples and 13,961 annotated images across six product categories (e.g., body shaping, height growth, health supplements) and includes two core tasks: Single-Violation (assessing fine-grained reasoning with short prompts) and All-in-One (testing long-context reasoning with unified, rule-intensive instructions). Through evaluating 26 mainstream LLMs and VLMs, the paper finds that state-of-the-art models frequently misclassify evasive content, and highlights the effectiveness of rule categorization, Retriever-Augmented Generation (RAG), and Supervised Fine-Tuning (SFT) in improving model performance. The work aims to lay the groundwork for safer e-commerce content moderation systems.

**Strengths:**

1. Working on a real-world application challenge by focusing on evasive content detection.
2. Constructs a large-scale, expert-annotated multimodal dataset (covering text and image inputs) with rigorous annotation pipelines, ensuring high-quality and regulation-aligned ground truth.
3. Conducts extensive experiments on 26 LLMs/VLMs (both open-source and closed-source), offering systematic baselines and detailed error analysis that sheds light on model limitations in multimodal reasoning.
4. Validates practical improvement strategies (RAG, SFT, rule merging) that can be directly applied to enhance real-world content moderation systems.

**Weaknesses:**

1. Unclear definition of Evasive Content Detection (ECD): The paper fails to provide a rigorous, cited definition of ECD, leaving ambiguity about its scope and distinguishing features. It also does not adequately justify why ECD is a unique and critical task for LLMs/VLMs, nor does it clarify how ECD differs from existing QA or adversarial detection tasks.
2. Narrow focus on domain-specific scenarios without generalizable capability assessment: The benchmark is limited to six e-commerce product categories, and the paper does not attempt to decouple domain-specific knowledge from core reasoning abilities (e.g., semantic understanding, logical inference). This limits the generalizability of the results to other evasive content detection scenarios.
3. Lack of novelty in research contributions: The work primarily reports experimental results using existing techniques (benchmark construction, RAG, SFT) without introducing innovative methods, frameworks, or theoretical insights. It functions more as a benchmark report than a novel research contribution.
4. Insufficient analysis of cross-category and cross-modality consistency: The paper does not deeply explore why model performance varies drastically across product categories (e.g., health supplements vs. body shaping) or modalities (text vs. image), missing opportunities to identify modality-specific or category-specific bottlenecks.
5. Limited discussion of benchmark generalizability and external validity: There is no analysis of whether EVADE-Bench’s samples are representative of real-world evasive content at scale, nor is there evidence that model performance on this benchmark correlates with real-world content moderation effectiveness.
6. Please fix grammer issue: english -> English, chinese -> Chinese

**Questions:**

1. How do the evasive content strategies in EVADE-Bench compare to those observed in non-e-commerce domains (e.g., social media, healthcare), and would the benchmark’s evaluation framework be adaptable to these domains?
2. Did the authors consider cross-lingual generalizability? Given that EVADE-Bench is Chinese-focused, would the observed model limitations and improvement strategies transfer to other languages?
3. How do the expert annotations account for evolving e-commerce regulations? Is there a plan to update the benchmark as policies change over time?
4. This paper mentioned adversarial attacks. What is the relationship between ECD and adversarial attacks? How can this benchmark help to defend against adversarial attacks?

**Details Of Ethics Concerns:**

The benchmark involves e-commercial content crawled from the web.

---

> ### Author Response · Authors · 2025-11-21
> **[Part 1/2] Response to Reviewer kXw3**
>
> We sincerely thank the reviewer for the insightful comments. We appreciate the opportunity to clarify the definitions, deepen our theoretical analysis, and highlight the scientific implications of our work. Below, we address your concerns and questions, incorporating new human baseline experiments and a formal complexity analysis to demonstrate the generalizability and depth of EVADE-Bench.
>
> ***
>
> **Q3: Expert annotations and evolving regulations**
>
> Our dataset construction pipeline is designed for adaptability. EVADE-Bench currently focuses on Chinese and relies on official advertising laws. **Any amendments to these official laws imply far-reaching changes in e-commerce rules across the entire nation. We are fully prepared to update the Rule Prompts via our pipeline immediately upon such legal changes to ensure EVADE-Bench always matches the latest regulations.**
>
> We have also submitted the complete results and analysis of expert annotation data. Please refer to **Complete Human Baseline Results and Annotation Quality Analysis** in the global comment.
>
> ***
>
> **W3: Lack of novelty in research contributions**
>
> Beyond the dataset construction, our work offers some insights and contributions. First, we discovered the Strong-Model Bias, a counter-intuitive phenomenon where stronger models like GPT-4o perform worse on prompt adherence than weaker models in specific scenarios **(lines 432-441)**. Second, we propose a novel Human-Model Collaborative Pipeline **(Algorithms 1 & 2 in Appendix C)**. We observed that while models possess text reasoning capabilities comparable to humans, humans have far superior multimodal understanding. By combining these strengths, we ensured the high quality of EVADE-Bench annotations, achieving a Krippendorff's Alpha greater than 0.71 **(Q3: Expert annotations and evolving regulations)**. Third, we found that category overlap in multi-class classification tasks poses a greater challenge to large models than long-context understanding **(Section 5.2)**. Fourth, we propose the insight of decoupling multimodal reasoning. Since the core challenge of ECD lies in the semantic ambiguity of the input, we suggest separating the steps of seeing (image description) and reasoning (logic application) to enhance model capabilities, rather than relying on single-step integrated inference **(Section 5.5)**.
>
> To verify the effectiveness of the multimodal decoupling approach (Section 5.5), we conducted experiments using GPT-4o and Claude-3.7 specifically for converting images into pure text descriptions, and then employed different powerful LLMs to perform multi-class inference on the generated text alone, without access to the images themselves. **Please refer to "Experimental Validation of Two-Stage VLM Decomposition" in the global official comment.**
>
> ***
>
> **W1 & Q4: Definition of Evasive Content Detection (ECD) and Relation to Adversarial Attacks**
>
> We apologize for the lack of clarity and will formalize the definition in our revision. ECD is fundamentally a restoration-then-reasoning task. In traditional QA tasks, the challenge lies in deriving the correct logical path from clear context to solve a problem. In contrast, **ECD requires models to first restore the original meaning from inputs that are deliberately obfuscated (e.g., via homophones, visual occlusion, or metaphors) before they can proceed with logical reasoning. This distinguishes it from traditional QA tasks where the input is generally unambiguous.**
>
> Furthermore, ECD differs from adversarial attacks like jailbreaking. While adversarial attacks aim to compromise model safety to generate harmful output, ECD acts at the data level with the goal of deceiving the model into misclassifying illicit content as compliant to bypass filters.
>
> ***
>
> **W4: Insufficient analysis of cross-category and cross-modality consistency**
>
> We have expanded our analysis using the Four-Level Rule System described in Appendix H and a newly conducted Human Baseline experiment. Regarding category inconsistency, the drastic performance drop in Health Supplements compared to Body Shaping is driven by logical depth. Body Shaping relies on Level 1 and 2 rules (Keyword and Pattern Matching), whereas Health Supplements require Level 3 and 4 rules (Combinatorial Logic and Exemption). Our analysis shows that current LLMs struggle exponentially as the rule hierarchy deepens. Regarding modality inconsistency, we observed an Inverse Gap when comparing models against our human baseline. On text tasks, SOTA models match human performance, but on image tasks, humans significantly outperform models with a full accuracy gap exceeding 20%. **This confirms that while humans naturally decipher visual metaphors, models suffer from a severe grounding failure and remain heavily text-dependent.**

---

> ### Author Response · Authors · 2025-11-21
> **[Part 2/2] Response to Reviewer kXw3**
>
> **Q1 & W2: Narrow focus on domain-specific scenarios and Comparison to non-e-commerce domains**
>
> We respectfully argue that while our data is sourced from e-commerce, our fundamental focus is on analyzing the reasoning boundaries of general-purpose LLMs and VLMs in ECD tasks. The benchmark serves to explore the performance and error sources of these general models when facing evasive samples. **If a foundation model fails to ground these abstract visual cues into logical rules here, it exposes a fundamental weakness in its robustness against semantic noise, a capability critical for any general-purpose AI deployment.** Additionally, our RAG and SFT experiments effectively decoupled domain knowledge from reasoning capabilities. The results show that even with injected domain knowledge, models still struggle with complex categories, proving that the bottleneck is reasoning complexity rather than domain specificity.
>
> **We confirm that similar data and benchmarks do exist in other fields, which we have reviewed in Section 2 (Related Work) of our paper to contextualize our contributions.**
>
> 1. Existence of Data: We acknowledge that data featuring "evasive" or "obfuscated" content is prevalent in non-e-commerce domains, particularly in Hate Speech and Cyberbullying . For instance, existing datasets address "lexical attacks"[4] and deceptive patterns in cyberbullying that require correction mechanisms[1]. Furthermore, research on "implicit hate detection"[5] utilizes autoregressive models to craft graded adversarial examples that are nuanced and context-sensitive, similar to the evasive nature found in our domain.
>
> 2. Existence of Benchmarks: Several multimodal benchmarks exist to evaluate robustness and safety in these broader contexts. The Hateful Memes Challenge[2] is a pioneering benchmark that evaluates multimodal reasoning using subtle hateful content contrasted with benign distractors . In the domain of general model safety, MM-SafetyBench[3] utilizes 5,040 adversarial image-text pairs to test vulnerability to malicious prompts , while MMSafeAware[7] evaluates safety awareness across 29 distinct threats . Additionally, VLD-Bench[6] focuses on disinformation detection using news-image pairs.
>
> References:
>
> [1] Deep learning approaches for detecting adversarial cyberbullying and hate speech in social networks, IEEE 2024.
>
> [2] The hateful memes challenge: Detecting hate speech in multimodal memes, NIPS 2021.
>
> [3] Mm-safetybench: A benchmark for safety evaluation of multimodal large language models, ECCV 2024.
>
> [4] Swe2: Subword enriched and significant word emphasized framework for hate speech detection. CIKM 2020.
>
> [5] Playing the part of the sharp bully: Generating adversarial examples for implicit hate speech detection. ACL 2023.
>
> [6] Vldbench: Vision language models disinformation detection benchmark, arxiv 2025.
>
> [7] Can’t see the forest for the trees: Benchmarking multimodal safety awareness for multimodal llms, ACL 2025.
>
> ***
>
> **Q2: Cross-lingual generalizability**
>
> We are developing an English version but prioritized Chinese because it presents a Hard Mode for ECD due to its logographic nature. In the 2D structure of Chinese characters, evasive operations like complex character splitting and visual puns appear much more frequently than in alphabetic languages. Models that succeed on the semantic density of Chinese EVADE-Bench are likely to possess robust reasoning capabilities transferrable to other languages.
>
> ***
>
> **W5: Generalizability and external validity**
>
> EVADE-Bench possesses high external validity as it is derived entirely from real-world industrial samples rather than synthetic data. These samples represent successful evasions collected from live platforms, meaning they have already bypassed existing industrial filters. Therefore, performance on this benchmark is a direct proxy for a model's capability to handle real-world, high-stakes moderation tasks involving naturally occurring adversarial data.
>
> ***
>
> **W6: Grammar issues**
>
> Thank you for pointing this out. We have thoroughly proofread the manuscript and corrected all grammatical errors, including the capitalization of English and Chinese. We will fix all similar grammatical issues in the next version of the PDF.

---

### Official Review · Reviewer_XC9t · 2025-11-01

**Soundness:** 3
**Presentation:** 3
**Contribution:** 4
**Rating:** 8
**Confidence:** 4

**Summary:**

This paper presents EVADE-Bench, a new Chinese multimodal benchmark designed to evaluate large language models (LLMs) and vision–language models (VLMs) on evasive content detection in e-commerce. The dataset contains 2,833 expert-annotated text samples and 13,961 annotated images across six high-risk product categories. The authors benchmark 26 open- and closed-source LLMs/VLMs, revealing substantial performance gaps and common reasoning failures. Additional experiments using retrieval-augmented generation (RAG) and supervised fine-tuning (SFT) demonstrate measurable improvements in detection accuracy. The paper further provides a detailed analysis of common error types and discusses a potential multi-agent or decomposed multimodal reasoning paradigm for future development. Overall, this work establishes the first rigorous benchmark standard for evaluating evasive content detection, offering a valuable foundation for developing safer and more transparent content moderation systems in e-commerce, and exploring effective strategies such as RAG and SFT to enhance large models’ compliance detection capabilities.

**Strengths:**

1. According to the authors, this work presents the first Chinese multimodal benchmark for evasive content detection in e-commerce scenarios. The study holds strong practical significance in detecting potentially deceptive or policy-violating advertisements and demonstrates both social value and pioneering potential in improving content safety and regulation in online commerce.

2. The authors employ a data construction pipeline that integrates human experts and large language models (LLMs), ensuring both the difficulty and quality of the dataset’s questions.

3. The paper conducts extensive experiments on the proposed benchmark, covering both open-source and closed-source models, as well as LLMs and VLMs, thoroughly validating the dataset’s effectiveness and quality.

4. In addition to the construction and evaluation of the dataset, this paper also examines the performance of RAG and SFT in optimizing the effectiveness of content avoidance detection, providing valuable insights and clear directions for further utilization of LLMs and VLMs in this field.

5. The authors provide comprehensive supplementary materials, including dataset construction methods, examples, and detailed definitions, which is commendable and enhances the transparency and reproducibility of the work.

**Weaknesses:**

1. The differences between the two task paradigms, Single-Violation and All-in-One, are not sufficiently explained.

2. The paper lacks an evaluation of annotation quality. Given that the task is a complex multi-class classification problem, inter-annotator agreement among experts is essential to ensure data reliability.

3. In the All-in-One task, the authors introduce two subtasks, “simplified description” and “detailed description,” but the motivation behind this design and the performance analysis across these subtasks are not adequately discussed.

4. The study does not include accuracy baselines from traditional methods (e.g., rule-based systems or human review), making it difficult to assess the true performance level of LLMs and VLMs on this task.

**Questions:**

1. Could the authors more clearly explain the conceptual and functional differences between the “single-violation” setting and the “all-in-one” setting? If possible, including detailed case examples in the supplementary material would better illustrate the differences and motivations behind these two settings.

2. The rationale for the partial-accuracy metric needs further explanation and analysis, especially in practical application scenarios. If a model tends to predict many violation categories, its partial accuracy could be very high — but this circumstance may be risky in real-world use.

3. Have the authors considered including human performance (from laypeople or experts) or traditional detection methods on the dataset? Given that evading content detection is a difficult task with many classes and hidden content, adding human accuracy would provide a better benchmark for model performance.

4. Regarding the particularly poor model performances in Table 2, did the authors verify that model inference was executed correctly? The authors suggest differences in model architecture and training strategies as an explanation — is this a reasonable account? Please provide detailed descriptions of how Deepseek-VL2 differs in architecture and training strategy from the other VLMs listed in the table.

5. The setups in Table 2 and Table 4 are confusing: Table 2 lists 23 models while Table 4 lists only 22. It appears that Deepseek-VL2, which performs poorly in Table 2, was omitted from Table 4 — this further raises doubts about whether Deepseek-VL2 was properly evaluated. The formats of Table 2 and Table 4 also need to be made consistent so readers can better compare model performance across tasks; consider uniformly distinguishing open-source and closed-source models.

6. From the tables, the authors only include the “simplified description” and “detailed description” sub-tasks under the All-in-One task — what motivated this choice? Also, please provide example cases in the supplementary material to demonstrate the specific differences between simplified and detailed descriptions.

7. In the error analysis section, the authors mention that stronger models are more likely to misinterpret prompt rules. One hypothesis is that stronger models possess more powerful reasoning capabilities and thus amplify deviations from human understanding — what is the detailed causal relationship here? Please supplement with a thorough explanation.

---

> ### Author Response · Authors · 2025-11-21
> **[Part 1/2] Response to Reviewer XC9t**
>
> We sincerely thank you for your positive assessment and the constructive feedback regarding the evaluation of annotation quality and the clarification of task settings. We have addressed your questions point-by-point below.
>
>
> **Q3 & W2 & W4: Human Annotation Quality & Baselines**
>
> In our original submission, we ensured annotation quality through a "Model-Human Collaborative Annotation" pipeline (Algorithm 2 in Appendix C), where models served as consistency checkers to flag ambiguous samples for expert re-verification. We initially believed this rigorous cross-check guaranteed the reliability of our Ground Truth (GT).
>
> However, inspired by reviewers' comments, we recognize the necessity of explicitly quantifying human performance to establish a baseline and verify the task's difficulty. **We conducted a Human Baseline Experiment involving three independent domain experts.**
>
> We have submitted the complete results and analysis of human annotation data. Please refer to **"Complete Human Baseline Results and Annotation Quality Analysis"** in the global official comment.

---

> ### Author Response · Authors · 2025-11-21
> **[Part 2/2] Response to Reviewer XC9t**
>
> **Q1 & W1: Difference between "Single-Violation" and "All-in-One"**
>
> The two tasks evaluate fundamentally different capabilities, distinguished by Ambiguity vs. Context Length:
>
> + Single-Violation (Fine-grained & Intra-Category Ambiguity): This task focuses on a single domain (e.g., Weight Loss). **The core challenge here is semantic overlap between fine-grained sub-categories**. For instance, distinguishing between "Exaggerated Weight Loss Claims" and "Direct Weight Loss Claims" involves subjective boundary definitions, or determining the line between "Direct" and "Implicit" claims. Our experiments show that this intra-category ambiguity significantly impacts model performance, serving as the primary difficulty factor in this setting.
> + All-in-One (Context Pressure & Disambiguation): This task addresses the ambiguity issue of Single-Violation by merging overlapping sub-categories (e.g., combining Exaggerated and Direct claims into a unified label). **While this reduces confusion during classification, it introduces a new challenge: Long-Context Pressure**. The model must process definitions for all six categories (7k+ tokens) simultaneously, testing its robustness in handling extensive instructions without losing focus.
> + Conclusion: We find that **semantic confusion (Single-Violation) often hinders models more than context length (All-in-One)**, as evidenced by the performance gains when sub-categories are merged. This is further supported by the fact that even with increased context length pressure, the same model consistently achieves significantly higher performance on All-in-One than on Single-Violation.
>
> **Q2: Rationale for Partial Accuracy**
>
> We agree that Partial Accuracy could theoretically be "gamed." Therefore, it is crucial to analyze it alongside Full Accuracy:
>
> 1. Recall vs. Precision: In content moderation, **Recall (approximated by Partial Accuracy) is often prioritized to minimize liability.**
> 2. Reward Hacking Detection: **Full Accuracy acts as a safeguard**. If a model achieves high Partial Accuracy but very low Full Accuracy, it indicates the model is attempting "reward hacking" by indiscriminately predicting multiple labels (over-recall) to ensure a "hit." Conversely, a narrower gap between the two metrics suggests genuine precision. We report both to provide a holistic view of model reliability.
>
> **Q4 & Q5: DeepSeek-VL2 & Table Consistency**
>
> + Missing from Table 4: DeepSeek-VL2-27B has a strict context limit of 4,096 tokens. The All-in-One prompt exceeds 7,000 tokens. It was physically impossible to run this model on the All-in-One task without truncating the rules (which would invalidate the results), hence its exclusion. **We have described this reason in lines 210-212 of the paper.**
> + Performance: We verified the inference logs. DeepSeek-VL2's lower performance stems from its architecture being optimized for general VQA rather than dense, rule-heavy reasoning, and its significantly smaller parameter count compared to SOTA models like Qwen-VL-72B or GPT-4o.
> + Format: We will unify the formatting of Table 2 and 4 to clearly distinguish open/closed source models as suggested.
>
> **Q6 & W3: "Simplified" vs. "Detailed" Instructions**
>
> + Simplified Instruction (Generalization): In this setting, we provide the category definition with minimal keywords. The objective is to test the model's generalization capability—specifically, whether it can infer violations that fall outside the provided keyword list by reasoning from the core definition ("first principles"), rather than relying on simple keyword matching.
> + Detailed Instruction (Instruction Following): We include specific positive and negative examples alongside comprehensive keyword lists. This tests the model's instruction-following capability and its ability to align with specific legal nuances via in-context learning.
>
> **Q7: Why Stronger Models Misinterpret Rules (Model Bias)**
>
> The causality lies in Instruction Following vs. Prior Beliefs. Stronger models (like GPT-4o) have been extensively aligned (RLHF) on general safety and helpfulness data, giving them strong internal priors.
>
> In EVADE-Bench, our rules are grounded in specific Chinese Advertising Laws, which may conflict with a model's general training distribution (e.g., a "before/after" comparison might be safe in general web data but illegal in Chinese ads).
>
> + Result: Weaker models, lacking strong priors, tend to follow the prompt's rigid logic mechanically. Stronger models exhibit a "confidence bias"—they prioritize their internal generalized definitions of safety over the specific in-context constraints, leading to the "Model Bias" errors we observed.
>
> We hope these clarifications and the new Human Baseline data satisfactorily address your concerns.

---

### Official Review · Reviewer_YKj2 · 2025-11-02

**Soundness:** 3
**Presentation:** 3
**Contribution:** 3
**Rating:** 6
**Confidence:** 4

**Summary:**

In this paper, the authors present a Chinese multimodal benchmark for evasive content detection for E-Commerce scenarios. The benchmark can support two kinds of evalution tasks: single violation and all-in-one, which help discriminate the reasoning ability under different fine-grained levels. The authors further perform evaluation for 26 open- and closed-source LLMs and VLMs under zero-shot, few-shot, RAG and SFT settings.

**Strengths:**

1. The paper is well written and easy to follow.
2. The task of evasive content detection is meaningful for many real-world scenarios. The proposed benchmark bears significant value towards the enhancement of the relevant techniques.
3. The authors perform a series of evaluation with the existing LLMs and VLMs. The results and error analysis shed a light on the further development.

**Weaknesses:**

1. The human annotation quality is not uncovered by the authors. It is unknown how hard this task is even for a human being. Also, fine-grained breakdown of the category distribution of the dataset should be reported.

**Questions:**

See the above 'Weaknesses'.

---

> ### Author Response · Authors · 2025-11-21
> **Response to Reviewer YKj2**
>
> We sincerely thank the reviewer for the constructive feedback. We appreciate the opportunity to clarify the quality of our human annotations and the breakdown of category distributions.
>
> **1. Response to "The human annotation quality is not uncovered"**
>
> In our original submission, we ensured annotation quality through a "Model-Human Collaborative Annotation" pipeline (Algorithm 2 in Appendix C), where models served as consistency checkers to flag ambiguous samples for expert re-verification. We initially believed this rigorous cross-check guaranteed the reliability of our Ground-Truth.
>
> However, inspired by reviewers' comments, we recognize the necessity of explicitly quantifying human performance to establish a baseline and verify the task's difficulty. **We conducted a Human Baseline Experiment involving three independent domain experts.**
>
> We have submitted the complete results and analysis of human annotation data. Please refer to **"Complete Human Baseline Results and Annotation Quality Analysis"** in the global official comment.
>
> ***
>
> **2. Response to "Fine-grained breakdown of the category distribution"**
>
> We apologize if the distribution details were not sufficiently prominent. **The breakdown of text/image counts and prompt lengths for all 6 categories is provided in Table 1 (Page 4) of the main paper**, and we will generate a fine-grained distribution chart based on data types and ground-truth distribution (e.g., count of "Rule A" vs. "Rule C"), **which will be included in the next PDF version** to provide a more granular view of the dataset's diversity and address your concerns regarding the robustness of our dataset construction.

---

### Author Response · Authors · 2025-11-22
**Complete Human Baseline Results and Annotation Quality Analysis**

We sincerely thank all reviewers for their valuable feedback, particularly regarding the need for a human baseline and quality analysis of human annotations. Due to the tight rebuttal timeline, we shared preliminary results for the 50% most difficult data under each reviewer's comments yesterday. We have now completed human annotation for the entire dataset, including inter-annotator agreement analysis. Please allow us to present the complete experimental results here.

**We conducted a Human Baseline Experiment with three independent domain experts to annotate all texts and images from EVADE-Bench.** To directly compare results and clearly show human annotation accuracy, we selected the top-performing open-source and closed-source models as benchmarks: DeepSeek-R1 and GPT-4.1-0414 for LLMs, and Qwen-VL-72B and Claude-3.7 for VLMs. **These models demonstrated the strongest overall performance. We also calculated Krippendorff's Alpha to analyze human annotation quality.**

*Note: Body. = Body Shaping, Women. = Women's Health, Height. = Height Growth, Men. = Men's Health, Weight. = Weight Loss, Health. = Health Supplement*

**Table-A: Human Performance on Image Split of EVADE-Bench (Human vs. SOTA Models, values are presented as Partial Accuracy / Full Accuracy).**

| | **Overall** | **Body.** | **Women.** | **Height.** | **Men.** | **Weight.** | **Health.** |
|:---|:---:|:---:|:---:|:---:|:---:|:---:|:---:|
| Qwen2.5-VL-72B | 57.63 / 26.05 | 57.78 / 20.81 | 61.93 / 42.08 | 67.70 / 26.64 | 55.87 / 32.11 | 72.57 / 02.99 | 44.35 / 27.38 |
| Claude-3.7 | 58.79 / 23.42 | **75.59** / 29.57 | 55.75 / 35.68 | 66.68 / 19.61 | 52.70 / 32.91 | 73.65 / 03.24 | 42.88 / 21.48 |
| Human on Image | **69.20** / **51.41** | 49.80 / **31.40** | **62.13** / **48.22** | **69.14** / **49.16** | **78.00** / **65.33** | **79.17** / **26.22** | **74.83** / **65.97** |

**Table-B: Human Performance on Text Split of EVADE-Bench (Human vs. SOTA Models, values are presented as Partial Accuracy / Full Accuracy).**

| | **Overall** | **Body.** | **Women.** | **Height.** | **Men.** | **Weight.** | **Health.** |
|:---|:---:|:---:|:---:|:---:|:---:|:---:|:---:|
| DeepSeek-R1 | 54.64 / 25.45 | **80.69** / **71.78** | 78.20 / 40.28 | 44.12 / 15.37 | 54.91 / 33.44 | 79.19 / 19.23 | 34.67 / 13.32 |
| GPT-4.1-0414 | 52.74 / 31.59 | 61.88 / 52.48 | 72.99 / 54.98 | 43.22 / 20.98 | 58.90 / 46.32 | 71.72 / 24.66 | 35.58 / 18.89 |
| Human on Text | **69.34** / **57.03** | 65.60 / 59.61 | **82.46** / **78.20** | **48.82** / **38.20** | **77.60** / **69.51** | **81.89** / **53.88** | **67.23** / **55.31** |

**Table-C: Human Annotation Quality Measured by Krippendorff's Alpha.**

|       | **Overall** | **Body.** | **Women.** | **Height.** | **Men.** | **Weight.** | **Health.** |
|-------|-------------|-----------|------------|-------------|----------|-------------|-------------|
| Human on Image | 0.7513      | 0.5961    | 0.5737     | 0.7638      | 0.6903   | 0.6502      | 0.7148      |
| Human on Text  | 0.6867      | 0.2472    | 0.3810     | 0.6137      | 0.6716   | 0.6719      | 0.6125      |

The results on the full dataset are consistent with our previous analysis of the hardest 50% of data.

- **Reliable Annotation Quality:** The Krippendorff's Alpha scores (0.67~0.8) indicate substantial agreement among experts, particularly on images (≥0.75 overall). This proves that despite the task's complexity, the rules are well-defined and consistent for humans, dispelling concerns about label noise.

- **Superior Human Performance:** Although models perform competitively on certain categories, humans still outperform SOTA LLMs and VLMs overall in understanding evasive text and images (Partial Accuracy). Moreover, humans also surpass current models in precision (Full Accuracy). On EVADE-Bench's most challenging data (Health Supplement), humans show an even larger performance gap over SOTA models. This demonstrates that humans have strong generalization ability with evasive content, maintaining stable reasoning performance across different types of text and images, unlike LLMs and VLMs which show high accuracy fluctuations across data types.

We will include these quantitative comparison tables in the revised Section 5.1 to strengthen the empirical rigor.

---

### Author Response · Authors · 2025-11-22
**Experimental Validation of Two-Stage VLM Decomposition**

We sincerely thank the reviewer for pointing out that Section 5.5 (Decompose VLM Reasoning) lacks corresponding experiments to support our conclusions. This feedback gave us an opportunity to strengthen our paper. In the error analysis (Section 5.4), **we hypothesized that VLMs undergo multimodal post-training on top of LLMs, which brings multimodal capabilities but may weaken the text processing abilities of the base LLM.** Based on this, we proposed in Section 5 that multimodal reasoning could be decomposed into two stages: **VLM generates image descriptions as text, and then LLM performs reasoning on the text.**

During the rebuttal period, we conducted experiments on EVADE-Bench's test set to validate this hypothesis and the decomposition approach. We used GPT-4o and Claude-3.7 as image description models, paired with different LLMs as text reasoning models, to observe how this method handles evasive content on EVADE-Bench. **To compare with "single-stage non-decomposed pure VLM reasoning", we used GPT-4o and Claude-3.7's direct inference results on the test set as baselines.** Below is our performance comparison between single-stage non-decomposed VLM reasoning and two-stage decomposed VLM reasoning:

**Table-A — GPT4o as Image Describer (Partial Accuracy).**

| Stage                                | Model      | Overall | Body. | Weight. | Women. | Men. | Health. | Height. |
|--------------------------------------|------------|---------|-------|---------|--------|------|---------|---------|
| Baseline (single-stage)              | GPT4o      | 59.61   | **76.82** | 74.11   | 55.20  | 53.72 | 45.64   | 65.26   |
| Two-stage (image described by GPT4o) | GPT4.1     | **61.65** | 67.95 | 79.91 | **59.60** | **56.65** | 49.20 | 69.91 |
| Two-stage (image described by GPT4o) | Qwen3-235b | 61.30   | 67.05 | **82.59** | 56.80  | 54.52 | 48.47   | **71.22** |
| Two-stage (image described by GPT4o) | DeepSeek-R1 | 60.87   | 69.55 | 78.57 | 58.00  | 50.53 | **51.66** | 67.15 |


**Table-B — Claude-3.7 as Image Describer (Partial Accuracy).**

| Stage                                      | Model        | Overall | Body. | Weight. | Women. | Men. | Health. | Height. |
|--------------------------------------------|--------------|---------|-------|---------|--------|------|---------|---------|
| Baseline (single-stage)                    | Claude-3.7   | 59.07 | **73.86** | 71.75 | 53.44 | 49.47 | 46.01 | 68.27 |
| Two-stage (image described by Claude-3.7)  | GPT4.1       | **62.55** | 62.95 | 78.12 | **61.60** | **60.64** | 51.66 | **71.51** |
| Two-stage (image described by Claude-3.7)  | Qwen3-235b   | 61.19 | 62.27 | **80.36** | 57.20 | 53.46 | **51.90** | 70.93 |
| Two-stage (image described by Claude-3.7)  | DeepSeek-R1   | 59.86 | 65.68 | 76.79 | 58.80 | 50.00 | 51.29 | 66.57 |

- The results show that compared to single-stage multimodal reasoning, using the same VLM to describe images and then passing the descriptions to a dedicated LLM for reasoning leads to better performance. **This is not an isolated phenomenon. As the table shows, this performance improvement is consistent across different VLMs and LLMs.**

- The performance did not saturate, indicating that the "Description" stage itself remains challenging due to the adversarial nature of EVADE-Bench (e.g., subtle visual puns). This further validates the high difficulty and value of our benchmark as a testbed for future "Restoration" capabilities.

- Additionally, **the experiment validates our hypothesis** that integrating "description + reasoning" in a single multimodal inference can weaken the model's inherent reasoning performance, suggesting that multimodal post-training may compromise the base LLM's text reasoning capabilities.

We will include these quantitative comparison tables in the revised Section 5.5 to strengthen the empirical rigor.

---

### Author Response · Authors · 2025-11-26
**Global Response: Summary of New Experiments (Human Baseline & Decomposition) & Clarifications**

We thank Reviewers 9eFs, kXw3, YKj2, and XC9t for their constructive feedback, which we have fully addressed by conducting two major new experiments (Human Baseline & VLM Decomposition) and providing detailed clarifications below.

**To Reviewer 9eFs:**

* **Regarding the concern on "repackaged pipeline" and contribution:** We clarified that EVADE-Bench defines a foundational "Restoration-then-Reasoning" capability distinct from standard QA, validated by the **significant performance gap between humans and models** we discovered.

* **Regarding the lack of empirical depth (VLM -> LLM):** We conducted **new experiments validating the Two-Stage Decomposition**, proving that decoupling visual description from logical reasoning significantly boosts performance.

* **Regarding data sources and ethics:** We confirmed that all data originates from public commercial listings under Fair Use principles, with no Personally Identifiable Information involved.

**To Reviewer kXw3:**

* **Regarding the definition of Evasive Content Detection (ECD):** We formalized ECD as a task targeting "semantic obfuscation" (restoring meaning from ambiguity), distinguishing it from adversarial attacks which target safety via noise injection.

* **Regarding domain specificity and generalizability:** We explained that Chinese serves as a "Hard Mode" for logographic ambiguity, and the reasoning challenges (grounding abstract cues into rules) are domain-agnostic.

* **Regarding cross-category inconsistency:** We utilized our Four-Level Rule System to explain that performance drops are caused by increasing logical depth (Level 3/4 rules) rather than domain differences.

**To Reviewer YKj2:**

* **Regarding the unknown human annotation quality:** We completed a **Human Baseline Experiment** on the full dataset, achieving a Krippendorff's Alpha > 0.71, confirming high annotation reliability.

* **Regarding fine-grained category breakdown:** We pointed to the existing data in Table 1 of our paper and committed to adding granular rule-distribution charts in the revision.

**To Reviewer XC9t:**

* **Regarding the difference between Single-Violation and All-in-One:** We clarified that Single-Violation tests robustness against *semantic ambiguity*, while All-in-One tests robustness against *context pressure*.

* **Regarding annotation quality and baselines:** We provided the **Human Baseline results**, showing humans significantly outperform SOTA models on image tasks (~69% vs 57% Partial Accuracy).

* **Regarding the Partial Accuracy metric:** We explained its necessity for prioritizing Recall in moderation tasks and its role in detecting "reward hacking" when paired with Full Accuracy.

* **Regarding DeepSeek-VL2 and table consistency:** We clarified its exclusion from All-in-One was due to physical context limits (4k vs 7k) and committed to unifying the table formats.

We have incorporated these new experiments and clarifications into our detailed replies. We kindly invite you to check our responses and the new data. We look forward to your feedback.

---

### Meta-Review · Area_Chair_JLFb · 2025-12-25

**Summary:**

The submission introduces EVADE-Bench, a multimodal benchmark aimed at detecting evasive content within Chinese e-commerce platforms. The authors propose a dataset covering six product categories and evaluate various LLMs and VLMs. While the practical utility of the dataset is acknowledged, the consensus decision is to reject the paper. The primary rationale is that the work is viewed as a domain-specific application report rather than a fundamental contribution to representation learning. Additionally, significant concerns remain regarding the limited technical novelty and potential data compliance issues.

**Reviewer Concerns:**

The authors made a substantial effort during the rebuttal, providing human baselines and decomposition experiments to address questions about data quality and experimental depth. However, these additions do not resolve the fundamental criticisms raised by Reviewers 9eFs and kXw3. The core issue remains that the benchmark is narrowly focused on Chinese e-commerce, limiting its generalizability and scientific impact for a broad venue like ICLR. Furthermore, the "methodological contribution" is perceived as a recombination of existing pipelines rather than a novel algorithmic advance. Crucially, the ethics concerns flagged regarding data provenance, copyright, and GDPR compliance (Reviewer kXw3) pose a significant barrier that has not been definitively cleared.

**Reviewer Scores:**

While Reviewers XC9t (Score: 8) and YKj2 (Score: 6) appreciated the data collection effort and may maintain their positive outlook based on the rebuttal, Reviewers 9eFs (Score: 2) and kXw3 (Score: 4) are likely to maintain their low scores. Their concerns target the foundational scope and novelty of the work, which the supplementary experiments did not structurally change. The split in scores reflects a disagreement on value (utility vs. novelty), but the lack of broad novelty and the ethics flag support a rejection.

---

### Decision · Program_Chairs · 2026-01-26

Reject